# Bacterial protein domains with a novel Ig-like fold target human CEACAM receptors

Nina M van Sorge[1,‡] (iD), Daniel A Bonsor[2] (iD), Liwen Deng[3] (iD), Erik Lindahl[4] (iD), Verena Schmitt[5] (iD),
Mykola Lyndin[5,6] (iD), Alexej Schmidt[7] (iD), Olof R Nilsson[8], Jaime Brizuela[9], Elena Boero[1],
Eric J Sundberg[2,10] (iD), Jos A G van Strijp[1] (iD), Kelly S Doran[3,†] (iD), Bernhard B Singer[5,†] (iD),
Gunnar Lindahl[8,11,*] (iD) & Alex J McCarthy[1,9,**] (iD)

## Abstract

*Streptococcus agalactiae*, also known as group B *Streptococcus* (GBS), is the major cause of neonatal sepsis in humans. A critical step to infection is adhesion of bacteria to epithelial surfaces. GBS adhesins have been identified to bind extracellular matrix components and cellular receptors. However, several putative adhesins have no host binding partner characterised. We report here that surface-expressed β protein of GBS binds to human CEACAM1 and CEACAM5 receptors. A crystal structure of the complex showed that an IgSF domain in β represents a novel Ig-fold subtype called IgI3, in which unique features allow binding to CEACAM1. Bioinformatic assessment revealed that this newly identified IgI3 fold is not exclusively present in GBS but is predicted to be present in adhesins from other clinically important human pathogens. In agreement with this prediction, we found that CEACAM1 binds to an IgI3 domain found in an adhesin from a different streptococcal species. Overall, our results indicate that the IgI3 fold could provide a broadly applied mechanism for bacteria to target CEACAMs.

**Keywords** Adhesin; immunoglobulin superfamily; IgI; receptor; *Streptococcus agalactiae*

**Subject Categories** Cell Adhesion, Polarity & Cytoskeleton; Microbiology, Virology & Host Pathogen Interaction; Structural Biology

**The EMBO Journal (2021) 40: e106103**

## Introduction

The Gram-positive bacterium *Streptococcus agalactiae*, also known as group B *Streptococcus* (GBS), is a major cause of pneumonia, septicaemia and meningitis in human neonates (Seale *et al*, 2017; Hall *et al*, 2017). Despite public health interventions, GBS is estimated to cause 410,000 infant infections and 150,000 stillbirths per year (Seale *et al*, 2017). From a total of 10 different serotypes, GBS belonging to serotypes Ia, Ib, II, III and V are most commonly associated with disease cases worldwide (Edmond *et al*, 2012). Cellular adhesion to epithelial surfaces is the first critical step preceding infection with GBS and many other pathogens (Patras & Nizet, 2018). Bacterial colonisation is a multifactorial process that requires expression of adhesins that target extracellular matrix (ECM) constituents and/or host cell receptors (Pietrocola *et al*, 2018; Shabayek & Spellerberg, 2018). Accordingly, GBS express a diverse array of surface adhesins and the exact repertoire varies from strain to strain, reflective of high plasticity of the GBS genome (Tettelin *et al*, 2005; Chen, 2019; Gori *et al*, 2020). Consequently, individual GBS strains differ in the host receptors that can be targeted during colonisation. The adhesins likely function cooperatively to facilitate persistent contact between the bacterial cell and the host, promoting cellular adhesion and invasion.

GBS interacts with several ECM constituents and the molecular mechanisms underpinning these interactions are well defined. The adhesins currently known to interact with ECM components are the fibrinogen-binding adhesins Srr1, FsbA, FsbB and FsbC (Schubert

1   Department of Medical Microbiology, University Medical Center Utrecht, Utrecht University, Utrecht, The Netherlands
2   Institute of Human Virology, University of Maryland School of Medicine, University of Maryland, Baltimore, MD, USA
3   Department of Immunology & Microbiology, University of Colorado Anschutz Medical Campus, Aurora, CO, USA
4   Department of Biochemistry and Biophysics, Science for Life Laboratory, Stockholm University, Stockholm, Sweden
5   Institute of Anatomy, Medical Faculty, University Duisburg-Essen, Essen, Germany
6   Department of Pathology, Sumy State University, Sumy, Ukraine
7   Department of Medical Biosciences, Umeå University, Pathology, Umeå, Sweden
8   Department of Laboratory Medicine, Division of Medical Microbiology, Lund University, Lund, Sweden
9   Department of Infectious Disease, MRC Centre for Molecular Bacteriology & Infection, Imperial College London, London, UK
10  Department of Biochemistry, Emory University School of Medicine, Atlanta, GA, USA
11  Department of Chemistry, Division of Applied Microbiology, Lund University, Lund, Sweden
    *Corresponding author. Tel: +46 735 342255; E-mail: gunnar.lindahl@tmb.lth.se
    **Corresponding author. Tel: +44 20 7594 3868; E-mail: a.mccarthy@imperial.ac.uk
    †These authors contributed equally to this work
    ‡Present address: Department of Medical Microbiology,, Infection Prevention and Netherlands Reference Laboratory for Bacterial Meningitis, Amsterdam University
     Medical Center, University of Amsterdam, Amsterdam, The Netherlands

et al, 2002; Gutekunst et al, 2004; Seo et al, 2012; Buscetta et al, 2014), the keratin 4-binding adhesin Srr1 (Samen et al, 2007), the fibronectin-binding adhesins BsaB and SfbA (Jiang & Wessels, 2014; Mu et al, 2014), and the collagen-binding adhesin PilA (Banerjee et al, 2011). These interactions promote epithelial colonisation (Schubert et al, 2004; Samen et al, 2007; Sheen et al, 2011; Wang et al, 2014) as well as cellular invasion and/or invasive disease (Tenenbaum et al, 2005; van Sorge et al, 2009; Banerjee et al, 2011; Seo et al, 2012; Mu et al, 2014; Deng et al, 2019). Generally, our knowledge of the mechanisms that GBS utilises to directly adhere to host cells is limited (Bolduc & Madoff, 2007; Patras & Nizet, 2018; Pietrocola et al, 2018); α adhesin binds α1β1-integrin to promote epithelial cell internalisation (Bolduc & Madoff, 2007), BspC adhesin binds vimentin (Deng et al, 2019), and the BspA adhesin interacts with gp340, a mucin-like glycoprotein associated with the surface of mucosal tissues (Rego et al, 2016). However, the host receptor targets of several putative adhesins remain uncharacterised, including Rib, Sip, LrrG, HvgA and BibA proteins (Stålhammar-Carlemalm et al, 1993; Brodeur et al, 2000; Santi et al, 2007; Tazi et al, 2010). Consequently, it is expected that several GBS adhesin–host factor interactions remain uncharacterised. Their identification is important for development of a vaccine or anti-bacterial strategies that interfere with GBS mucosal colonisation (Michel et al, 1992; Larsson et al, 1996; Heath, 2016; Pietrocola et al, 2018).

We report here that a subset of GBS strains interact with carcinoembryonic antigen-related cell adhesion molecules (CEACAM) receptors. Human CEACAMs are a family of 12 cellular receptors belonging to the immunoglobulin (Ig) superfamily (IgSF) that are commonly expressed on epithelial cells, endothelial cells and leucocytes (Gray-Owen & Blumberg, 2006). CEACAMs regulate immune responses through formation of homophilic and heterophilic interactions (Kuespert et al, 2006; Gray-Owen & Blumberg, 2006; Bonsor et al, 2015b). These properties allow CEACAMs to regulate multiple physiological and pathophysiological processes including cell-to-cell communication, epithelial differentiation, apoptosis and regulation of pro-inflammatory reactions (Gray-Owen & Blumberg, 2006; Khairnar et al, 2015, 2018; Helfrich & Singer, 2019). Each CEACAM is composed of an N-terminal domain with V-set (IgV) fold, which promotes CEACAM-CEACAM interactions, and varying numbers of C2-set (IgC2) folds (Zhou et al, 1993; Watt et al, 2001; Bonsor et al, 2015a, 2015b). Interestingly, human CEACAMs have been identified as docking receptors for Gram-negative bacteria (Chen & Gotschlich, 1996; Virji et al, 1996; Hill et al, 2001; Conners et al, 2008; Tchoupa et al, 2014; Tchoupa et al, 2015; Javaheri et al, 2016; Königer et al, 2016; Brewer et al, 2019). The characterised bacterial adhesins include Opa of Neisseria spp., HopQ of Helicobacter pylori, UspA1 of Moraxella catarrhalis, P1 of Haemophilus influenzae, Dr adhesins of Escherichia coli and CbpF of Fusobacterium spp. Interaction of these adhesins with human CEACAMs promotes cellular adhesion, cellular invasion, translocation of virulence factors and tissue penetration (Billker et al, 2002; Korotkova et al, 2008; Tchoupa et al, 2014; Tchoupa et al, 2015; Königer et al, 2016; Javaheri et al, 2016; Islam et al, 2018). Of note, these CEACAM-binding bacterial adhesins are structurally distinct, implying that they arose through convergent evolution. Since CEACAMs have not previously been reported to interact with Gram-positive bacteria, it was therefore of considerable interest to characterise the mechanism by which GBS binds to CEACAM receptors.

We used biochemical and cellular assays to reveal that the GBS surface protein β binds specifically to human CEACAM1 and CEACAM5 and that an IgSF domain in β promotes the binding. Through structural methods, we demonstrate that the IgSF domain in β protein contains unique structural features and represents a previously unrecognised variant of the IgI fold, which we termed the I3-set fold (IgI3). Homologs of β-IgI3 were identified in adhesins from several human bacteria including pathogens. We confirmed that CEACAM1 recognised one of these IgI3 variants and two streptococcal species. Together, these findings suggest that the interaction between bacterial IgI3 domains and CEACAMs may be of general importance for host barrier colonisation.

## Results

### The group B *Streptococcus* surface β protein binds CEACAM1

GBS can successfully adhere to human epithelial cell lines, including vaginal, cervical and airway epithelial cell lines (Patras & Nizet, 2018). Yet, knowledge of the specific host factors that can be targeted by GBS for adhesion is limited. As CEACAM receptors are mucosal docking targets for microorganisms (Tchoupa et al, 2014), we hypothesised that GBS interacts with CEACAM receptors. Given that individual GBS strains are likely to vary in their molecular adhesion mechanisms due to genomic plasticity (Tettelin et al, 2005; Chen, 2019; Gori et al, 2020), we tested interaction of rCEACAM1 with a panel of genetically diverse genome-sequenced GBS strains. We used CEACAM1 in the screen as it binds all known bacterial ligands of CEACAMs. Using rCEACAM1 (Fig EV1A), we observed binding to GBS strains including reference strains A909 and H36B (Figs 1A and EV1A). To further test the specificity of the interaction, we also screened the ability of rCEACAM1 to interact with a variety of other Gram-positive species including Streptococcus spp., Enterococcus spp. and Staphylococcus spp. Notably, rCEACAM1 only bound to the indicated GBS strains (Fig 1A). These data reveal for the first time that GBS can bind to human CEACAM1 receptor.

Next, we aimed to identify the GBS adhesin responsible for binding rCEACAM1. We utilised the GBS genome comparison dataset of Tettelin et al. (2005), which includes rCEACAM1 binding (A909 and H36B) and non-binding (18RS21 and NCTC10/84) strains, to identify genes encoding cell wall-anchored proteins that were associated with rCEACAM1-binding phenotype. We identified that genomic island of diversity region 3.1 was present in rCEACAM1-binding strains (A909 and H36B) and absent in non-binding strains (515, COH1, NEM316 and NCTC10/84). This region encodes a cell wall-anchored protein known as β protein, or simply β, that is encoded by bac. To test the hypothesis that β protein was responsible for CEACAM1 interaction, we screened rCEACAM1 binding to a broader collection of GBS isolates (Fig 1B) and observed that CEACAM1 binding correlated with carriage of bac (Hedén et al, 1991; Lindahl et al, 2005). The intensity of rCEACAM1 binding differed between interacting GBS strains, which may reflect differences in β expression. Indeed, β protein expression levels, detected using anti-β serum (Fig EV1B), correlated with rCEACAM1-binding capacity (Fig 1C). Finally, we tested whether CEACAM1 binding was dependent on β protein expression. Deletion of bac in the GBS A909

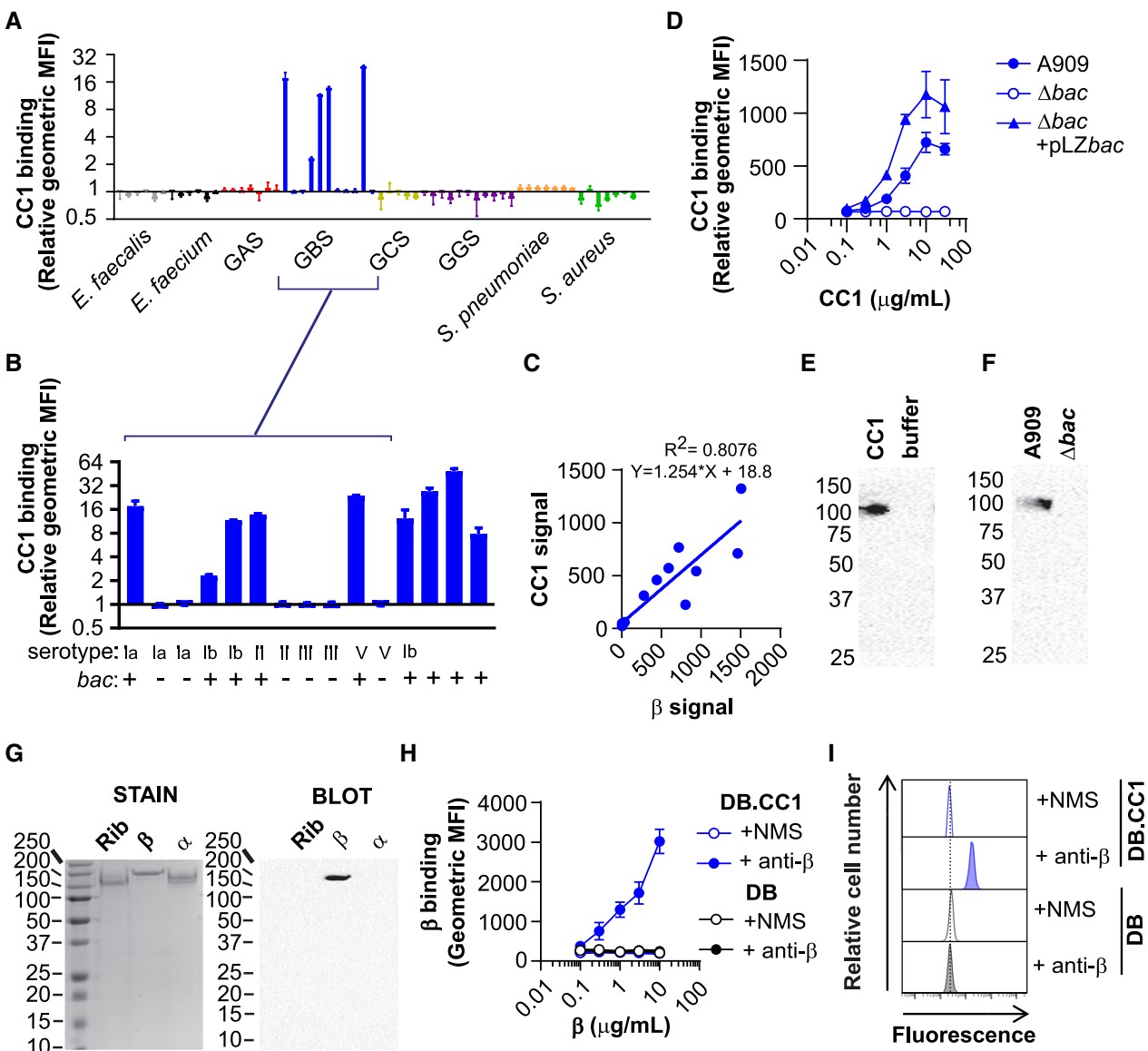

**Figure 1.** Human CEACAM1 binds to group B *Streptococcus* (GBS) via the surface-expressed β protein.

A, B  Binding of recombinant (r)CEACAM1 (CC1)-HIS (10 μg/ml) to (A) a panel of Gram-positive bacteria, namely *Enterococcus faecalis*, *Enterococcus faecium*, group A *Streptococcus* (GAS), group B *Streptococcus* (GBS), group C *Streptococcus* (GCS), group G *Streptococcus* (GGS), *Streptococcus pneumoniae*, *Staphylococcus aureus*, (B) an expanded collection of GBS isolates, with serotype and carriage of *bac* gene shown. rCC1 binding to live bacteria was quantified by a secondary anti-His-FITC monoclonal antibody (mAb). Mean and standard deviation (SD) values are reported for *n* = 3 independent experiments. In (B), GBS strains from left to right are A909, BS39, 515, BS22, H36B, BS29, 18RS21, BM110, COH1, SBL3066, NCTC10/84, SB35, SB10, SB20, BS26.

C  Correlation between rCC1-binding capacity and β protein expression in GBS strains, respectively. β protein expression was quantified using rabbit anti-β protein serum and a secondary PE-conjugated goat anti-rabbit-IgG. rCC1 binding to live bacteria was quantified by a secondary anti-His-FITC mAb. Mean binding values from *n* = 3 independent experiments.

D  Concentration-dependent binding of rCC1-HIS to wild type (WT), Δ*bac* and Δ*bac* + pLZ.*bac* complemented A909 strains. Mean and SD values are reported for *n* = 3 independent flow cytometry experiments.

E  Pull-down of human rCC1-HIS (30 μg/ml) by GBS strain A909 strain, detected using Western blot analysis.

F  Human rCC1-HIS (30 μg/ml) binding to GBS A909 and A909Δ*bac* strains analysed by pull-down experiments followed by Western blot analysis.

G  Western blot analysis of rCC1-HIS (30 μg/ml) binding to purified β protein, but not to control GBS surface proteins α and Rib. Panel on the left show staining of proteins after separation by SDS-PAGE.

H, I  Binding of purified β protein to dynabeads (DB) coated with rCC1 (DB.CC1) or control. Binding of β protein was detected using mouse anti-β serum or normal mouse serum (NMS), and PE-conjugated goat anti-mouse-IgG. (H) shows mean and SD values compiled from *n* = 3 independent replicates, and (I) shows representative flow cytometry plots using 10 μg/ml rCC1.

Data information: Fluorescence of bacteria (A, B, C and D) and DB (H and I) was measured by flow cytometry. Western Blot in (E, F and G) employed mouse anti-CC1 mAb (clone C5-1X/8) and HRP-conjugated goat anti-mouse-IgG pAb.

reference strain abolished rCEACAM1 binding, whilst complementation with a vector carrying the *bac* gene re-established rCEACAM1 binding (Figs 1D and EV1C). Mutation of genes encoding other GBS surface adhesins or capsule did not influence interaction with rCEACAM1 (Fig EV1C). Additionally, we were able to pull down rCEACAM1 with GBS A909 (Fig 1E), but not the Δ*bac* strain (Fig 1F). Direct interaction of rCEACAM1 with purified β protein, but not with purified GBS surface proteins Rib or α, was demonstrated by Western blot analysis (Fig 1G). Finally, β protein bound to CEACAM1-coated dynabeads (DB), but not control beads, in a concentration-dependent manner (Fig 1H and I). Together, these data unequivocally show that GBS specifically binds to human CEACAM1 through expression of β protein.

### CEACAM1 binds to an IgSF domain in β protein

β protein is a multifunctional protein and interacts with several proteins of the human immune system, including factor H, Siglec-5, Siglec-7, Siglec-14 and IgA, through distinct domains (Fig 2A) (Lindahl *et al*, 1990; Hedén *et al*, 1991; Areschoug *et al*, 2002b; Carlin *et al*, 2009; Nordström *et al*, 2011; Ali *et al*, 2014; Fong *et al*, 2018). To pinpoint the β protein domain interacting with CEACAM1, we expressed and purified recombinant β protein domains (B6N, IgABR, B6C, IgSF and β75KN) from *E. coli* (Fig 2B) and tested their interaction with rCEACAM1. These domains include a part of β called IgSF because the domain was predicted to adopt an immunoglobulin superfamily (IgSF) fold (Bateman *et al*, 1996; Lindahl *et al*, 2005). IgSF folds are composed of around 100 amino acids comprising two β sheets that pack face-to-face. rCEACAM1 bound in a concentration-dependent manner to DB coated with the biotinylated IgSF domain, but not to DB coated with other β protein domains or biotinylated-HSA (Fig 2C). As a control, we included rSiglec-5, which interacted only with DB coated with the B6N domain as described previously (Fig 2C; Nordström *et al*, 2011). Furthermore, DB coated with rCEACAM1 interacted with the IgSF domain, but not HSA, coupled to streptavidin (Fig 2D). Thus, the IgSF domain in β specifically binds to human CEACAM1. We assessed whether the IgSF domain shared structural homology with other resolved IgSF structures. Structural predictions of the β-IgSF domain suggested resemblance to a V-set Ig (IgV) domain identified in SrpA, a cell surface-anchored protein

in *Streptococcus sanguinis* (Fig 2E; Bensing *et al*, 2016). However, rCEACAM1 did not bind to the surface of a SrpA-expressing *S. sanguinis* strain (Appendix Fig S1). We denoted the IgSF fold in β as β-IgSF.

The CEACAM1-4L isoform possesses four Ig-like domains (Appendix Fig S2A; Gray-Owen & Blumberg, 2006), including the N-terminal IgV-like domain that forms the homo- and heterophilic interactions implicated in the CEACAM-mediated functions (Bonsor *et al*, 2015b). All bacterial ligands characterised to date for CEACAM1 target the N-terminal IgV-like domain. To identify the CEACAM1 domain targeted by β-IgSF, we tested whether domain-specific CEACAM1 monoclonal antibodies (mAb) could inhibit the interaction of β-IgSF with CEACAM1-coated DB. Only a mAb that blocks the N-terminal domain of CEACAM1 abolished the interaction with β-IgSF (Fig 2F; Appendix Fig S2B). Similarly, the *H. pylori* protein HopQ, which binds to the N-terminal domain (Bonsor *et al*, 2018), blocked the β-IgSF-CEACAM1 interaction (Appendix Fig S2C). To test for direct interaction with the N-terminal domain, we coated DB with rCEACAM1-N or rCEACAM1-A1B1A2 (Appendix Fig S2D) and observed concentration-dependent binding of β-IgSF only to DB coated with rCEACAM1-N (Fig 2G). In the reciprocal experiment, rCEACAM1-N, but not rCEACAM1-A1B1A2, bound to DB coated with β-IgSF (Appendix Fig S2E).

### The β-IgSF domain binds with high affinity to human CEACAM1 and CEACAM5

Bacterial adhesins of CEACAM1 often also bind to the N-terminal domain of CEACAM3, CEACAM5 and CEACAM6, reflecting the high sequence (~90% identity) and structural homology between CEACAM family members (Popp *et al*, 1999; Gray-Owen & Blumberg, 2006). However, no bacterial adhesins to date bind CEACAM8. To ascertain the CEACAM-binding profile for β-IgSF, we measured the affinity of β-IgSF for unglycosylated N-terminal domains of these five CEACAMs, purified from *E. coli*, by isothermal calorimetry (ITC) (Fig 3A; Appendix Table S3). β-IgSF bound with high affinity to rCEACAM1-N ($K_D$ = 96 ± 2 nM) and rCEACAM5-N ($K_D$ = 152 ± 27 nM), but did not bind to rCEACAM3-N, rCEACAM6-N or rCEACAM8-N. The affinities of the (β-IgSF)-(CEACAM1-N) and (β-IgSF)-(CEACAM5-N) interactions were comparable to those reported for other bacterial ligands (Korotkova

---

**Figure 2. The IgSF domain in the β protein of GBS binds to the N-terminal domain of CEACAM1.** ▶

A  Schematic representation of the β protein (Hedén *et al*, 1991). The first residue of each domain is indicated. XPZ is a proline-rich repeat region (Areschoug *et al*, 2002a). Known human ligand binding partners are indicated above each respective domain. IgA binds to a region denoted IgABR. Factor H binds the β75K region, but does not bind via XPZ.

B  Staining of recombinant β protein domains (B6N, IgABR, B6C, β-IgSF and β75KN) after separation by SDS-PAGE. Recombinant proteins were expressed in *E. coli*. β protein purified from GBS is used as a control. Dotted boxes signal proteins of expected size.

C  Binding of rCC1-His or rSiglec5-Fc to dynabeads (DB) coated with β protein domains. Mean and SD values are reported for *n* = 2 replicates.

D  Binding of varying concentrations of β-IgSF or human serum albumin (HSA) coupled to streptavidin-PE (SPE), to DB coated with CC1 (DB.CC1) or control protein (DB). Mean and SD values are reported for *n* = 6 replicates.

E  Structure of the relevant (Siglec-like sialic acid-binding) domain of the highest-scoring template 5KIQ (SrpA binding region), and predicted three-dimensional model of the IgSF domain from the β protein, constructed from the Hhpred alignment using MODELLER (Webb & Sali, 2016).

F  Inhibition of β-IgSF binding to DB.CC1 by anti-CC1-N mAb (clone CC1/3/5-Sab), but not by anti-CC1-A1B1 mAb (clone B3-17). β-IgSF-biotin was coupled to SPE. Mean and SD values are reported for *n* = 6 replicates.

G  Concentration-dependent binding of β-IgSF to DB coated with 100 μg/ml CC1 (DB.CC1), CC1-N (DB.CC1-N) or CC1-A1B1A2 (DB.CC1-A1B1A2). Assays utilised proteins coupled to SPE. Mean and SD values are reported for *n* = 6 replicates.

Data information: Fluorescence of DB in (C, D and F) was measured by flow cytometry.

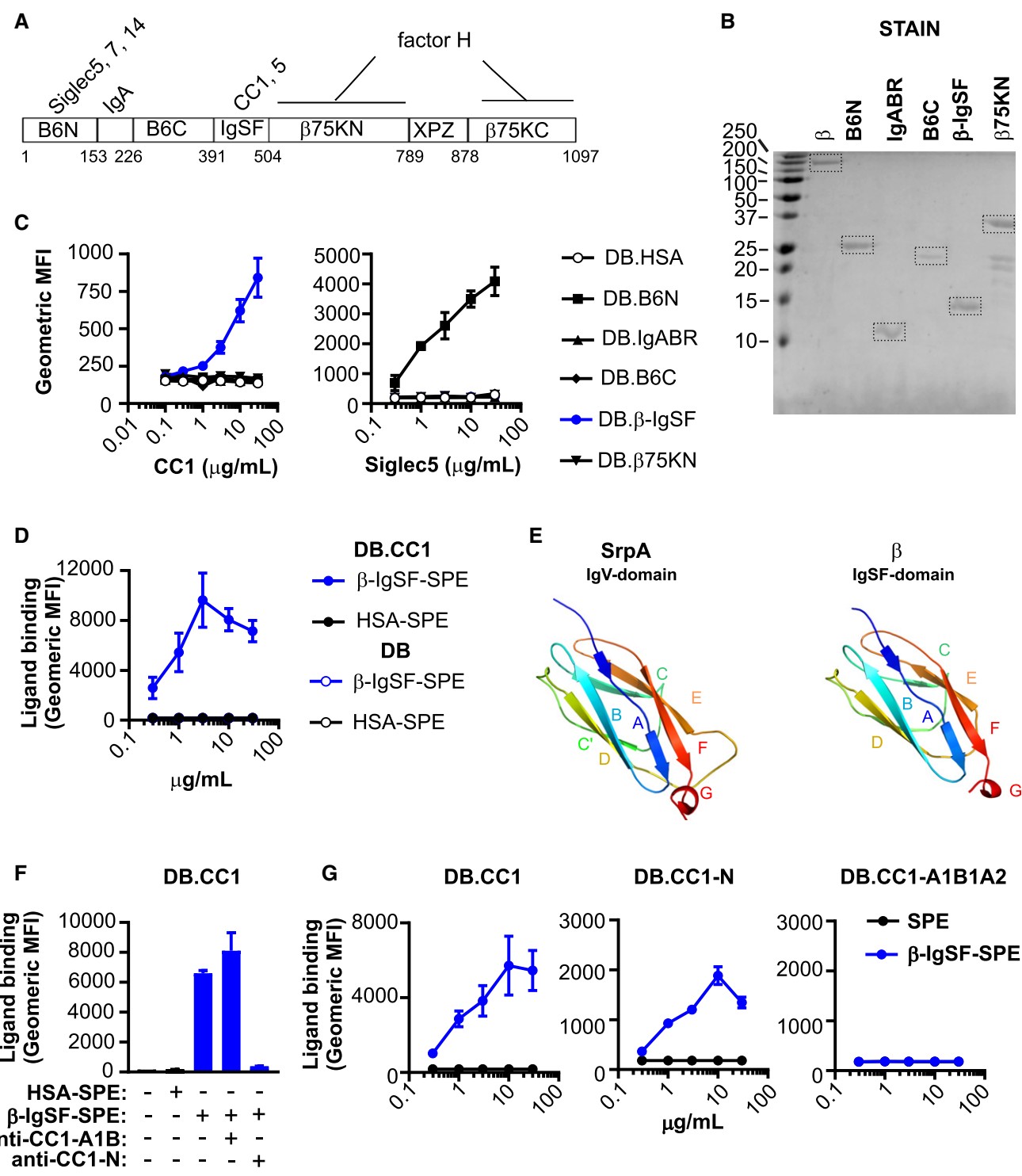

Figure 2.

et al, 2008; Bonsor et al, 2018). As the binding of β-IgSF is weaker to the isolated N-terminal domain compared to glycosylated CEACAM1 (Fig 2G), other CEACAM1 domains may stabilise the interaction, as reported for the HopQ-CEACAM1 interaction (Bonsor et al, 2018). To confirm that the CEACAM-binding profile of the protein is valid when expressed on the surface of GBS bacteria, we tested binding of the full-length glycosylated rCEACAMs to GBS by

flow cytometry. The β-expressing GBS A909 strain only recognised CEACAM1 and CEACAM5 (Fig 3B). Specificity for this subset was generally confirmed in a wider panel of β-expressing GBS strains (Appendix Fig S3), though variation existed in CEACAM5 binding. Although several β-expressing GBS strains did not bind CEACAM5, the β-IgSF domain displayed 100% sequence conservation amongst 57 GBS proteins identified through BLAST search (Appendix Fig

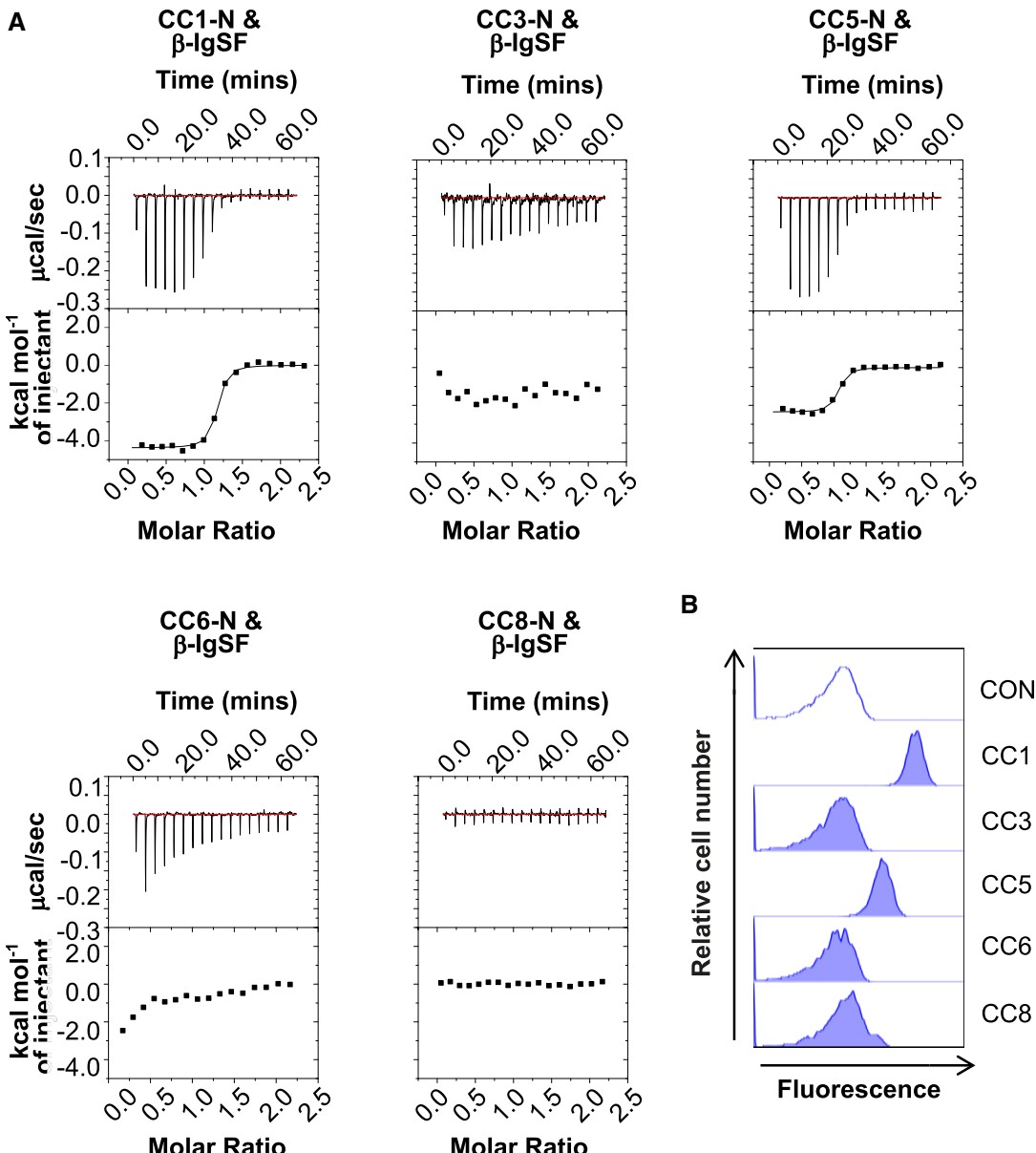

**Figure 3.  β-IgSF binds with high affinity to human CEACAM1 and CEACAM5.**

A   Representative isothermal titration calorimetry (ITC) binding curves of CEACAM-N domains and β-IgSF duplicates. Experiments were performed using β-IgSF and N domains of (r)CEACAM1 (CC1), CEACAM3 (CC3), CEACAM5 (CC5), CEACAM6 (CC6) and CEACAM8 (CC8).

B   Representative flow cytometry plots analysing the interaction of rCC1, CC3, CC5, CC6 or CC8 (10 μg/ml) with GBS A909. Fluorescence of bacteria was measured by flow cytometry. Representative data from $n = 3$ independent experiments.

S4), indicating that intrastrain variation in CEACAM5 binding is most likely due to differential β expression.

## Expression of human CEACAMs on epithelial cells enhances GBS adhesion

As CEACAM engagement leads to enhanced cellular adhesion of Gram-negative microorganisms, we hypothesised that CEACAMs could represent a novel cellular adhesion mechanism for GBS. Therefore, we tested the binding of strain A909 to well-characterised CEACAM-expressing HeLa cells (Fig 4A; Bos *et al*, 1998). Consistent with rCEACAM binding, a higher percentage of the A909 inoculum was recovered from the CEACAM1- and CEACAM5-expressing HeLa cells in comparison to all other HeLa cell lines (Fig 4A). Increased binding of GBS to human CEACAM1-expressing HeLa cells was confirmed by confocal microscopy (Fig 4B). To ensure that the CEACAM-observed binding was not influenced by the background of the cell line, we assessed GBS adhesion to CEACAM-expressing CHO cells (Hollandsworth *et al*, 2020). Also, a higher percentage of the GBS inoculum was

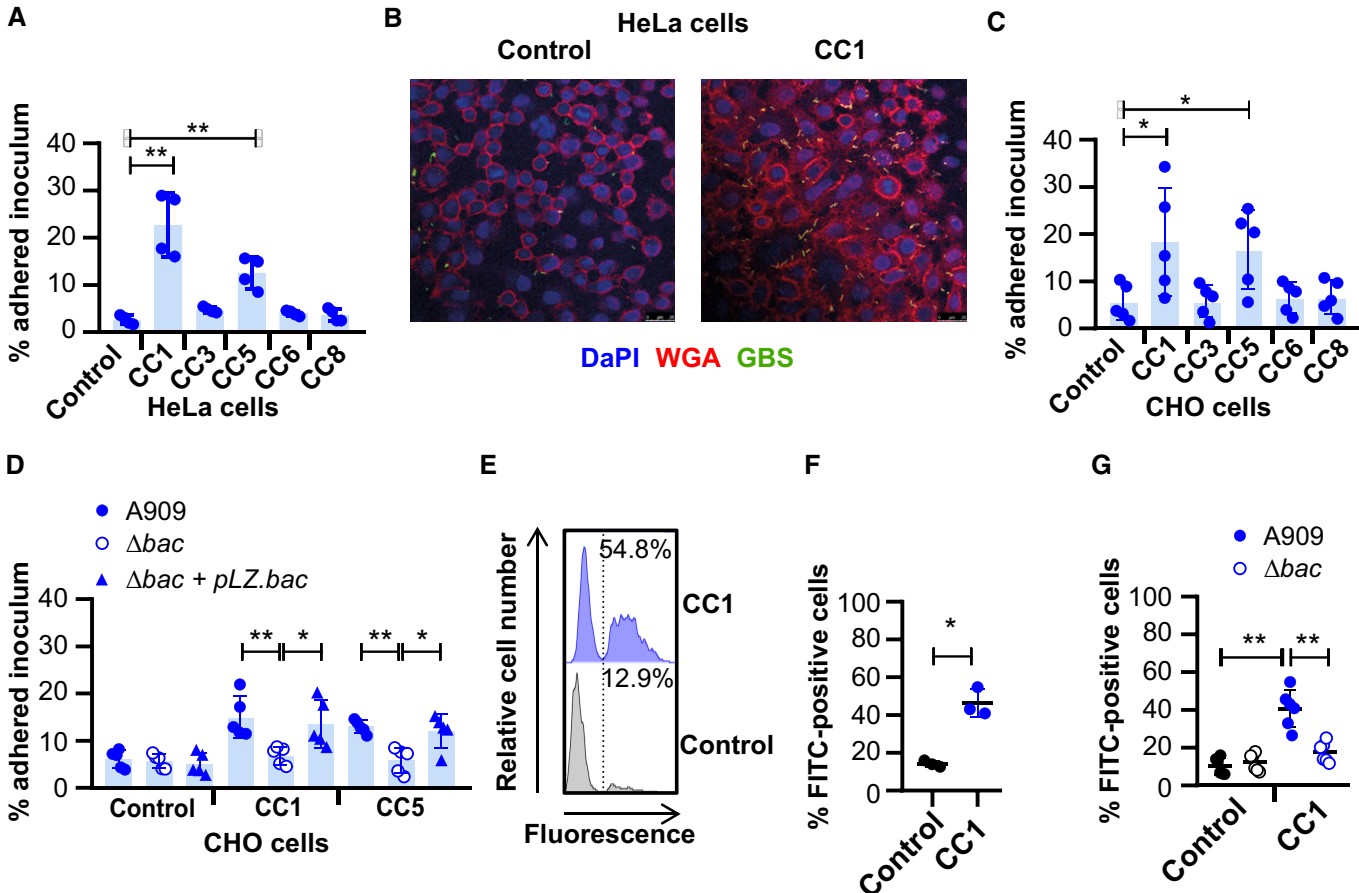

**Figure 4. GBS exploit human CEACAM for enhanced adhesion to epithelial cells.**

A   Adherence of GBS A909 to human CEACAM1 (CC1)-, CEACAM3 (CC3)-, CEACAM5 (CC5)-, CEACAM6 (CC6)- or CEACAM8 (CC8)-expressing, or an empty vector control, HeLa transfectants at MOI of 10 for 30 min. Mean and SD for $n = 4$ independent replicates are shown.

B   Confocal imaging of FITC-labelled GBS A909 strain adhesion to human CC1-expressing or empty vector control HeLa transfectants. Cell membranes were stained with AF-647-conjugated wheat germ agglutinin (WGA) (red) and nuclei with DAPI (blue). Scale bar: 26 μm.

C   Adherence of A909 to human CC1-, CC3-, CC5-, CC6- or CC8-expressing, or an empty vector control, CHO transfectants at MOI of 10 for 30 min. Mean and SD from $n = 5$ independent replicates is displayed.

D   Adhesion of WT, Δbac and Δbac + pLZ.bac complemented A909 strains to human CC1- or CC5-expressing CHO transfectants, or empty vector control cells. Mean and SD from $n = 5$ independent replicates is displayed.

E, F   Adhesion of FITC-labelled A909 to human CC1-expressing, or control, CHO transfectants at an MOI of 10. Cell lines were detached and then incubated with FITC-labelled A909 for 30 min at 4°C. The percentage of fluorescent cells was calculated for each population. Representative flow cytometry plots are shown (E), and the integrated results displaying the mean and SD from 3 independent replicates are reported in (F).

G   Adhesion of FITC-labelled A909 strains to human CC1-expressing or control CHO transfectants at MOI of 10. Cell lines were detached and then incubated with FITC-labelled A909 strains for 30 min at 4°C. Mean and SD for each population for $n = 6$ independent replicates is displayed.

Data information: Fluorescence of cells in (E, F and G) was measured by flow cytometry analysis, where the percentage of fluorescent cells was calculated. The number of bacteria adhered to cell monolayers in (A, C and D) were quantified by growth on Todd-Hewitt agar and enumeration of colony forming units (CFU). Data in (A, C, D, F and G) were analysed by repeated one-way ANOVA with Sidak's multiple comparisons. *$P < 0.05$ and **$P < 0.01$.

recovered from the CEACAM1- and CEACAM5-expressing CHO cells in comparison to all other CHO cell lines (Fig 4C). For unknown reasons, there was considerable variation in GBS adhesion to the CEACAM1- and CEACAM5-expressing CHO cells. This variation could not be attributed to unstable CEACAM expression, as our cell lines expressed CEACAM at consistent levels (Fig EV2A and B). It is possible that the variation in GBS adhesion could be due to differences in CEACAM-expression density by CHO cells or due to differences in β expression by GBS strains in replicated experiments.

To confirm that adhesion of the GBS strain A909 to the CEACAM1-expressing CHO cell line was dependent on β protein, we also assessed binding of the *bac* deletion mutant and the complemented strain in this same system. GBS adherence was abolished by mutation of the *bac* gene, and the phenotype could be recapitulated by the complemented mutant (Fig 4D). This result was confirmed by confocal microscopy (Fig EV2C). In agreement with the results obtained with purified proteins (Fig 2F), the binding of β-expressing GBS to CEACAM1-expressing CHO cells was inhibited by blocking the CEACAM1-N domain with a specific mAb (Fig EV2D).

Moreover, pre-incubation of A909 with rCEACAM1-N, but not rCEACAM8-N, impaired adhesion to CEACAM1-expressing CHO transfectants (Fig EV2E). This indicates that the CEACAM1-N and β-IgSF domains were both responsible for the cellular adhesion phenotype.

To rule out the possibility that differences in adhesion of A909 wild-type and Δ*bac* strains to CHO cells reflected interstrain variation in growth rates during adhesion at 37°C, we developed an alternative assay in which we assessed adhesion of FITC-labelled GBS strains to detached cell lines during incubation at 4°C. A higher percentage of CEACAM1-expressing CHO cells were FITC-positive upon incubation with A909 in comparison to control cells (Fig 4E and F). As expected, adhesion of FITC-labelled A909 to the CEACAM1-expressing CHO cell was abolished by mutation of *bac* (Figs 4G and Fig EV2F). Collectively, the data therefore indicate that binding to human CEACAM receptors, via β, is a novel cellular adhesion mechanism for GBS.

## Crystallography reveals that the IgSF domain in β adopts a novel Ig fold, the IgI3 fold

To gain insights into β-IgSF binding mechanisms, we aimed to solve its structure. We solved the structure in complex with CEACAM1-N at a resolution of 3.25 Å (Fig 5A). The asymmetric unit contains two molecules of the (β-IgSF)-(CEACAM1-N) complex, which are similar to each other when superimposed (r.m.s.d 0.20 Å). All analysis of the (β-IgSF)-(CEACAM1-N) complex has been performed with chains B and C of the co-crystal complex. The β-IgSF domain has the characteristic features of an Ig fold, principally a pair of β sheets built of anti-parallel β strands that surround a hydrophobic core (Fig 5B). IgSF domains can be classified into (variable) V-set, (constant) C-set or (intermediate) I-set, with differentiation based on the number and placement of β-strands between the conserved cysteine residue disulphide bridge (Fig 5C; Wang & Springer, 1998; Wang, 2013). The V-set Ig (IgV) domains contain ten β strands with four strands found on one sheet (*ABED*) and six strands on the other (*A'GFCC'C''*). C-set Ig (IgC) domains lack the *A'* and *C''* strands and are further grouped into C1 or C2 based on presence or absence of the *D* strand, respectively. I-set Ig (IgI) domains lack the *C''* strand and are classified into I1 or I2 based on presence or absence of a *D* strand, respectively. The β-IgSF domain has two β-sheets labelled *ABED* and *A'GFC*, with sheets connected by the *BC*, *EF*, *CD* and *AA'* loops (Fig 5B). Therefore, β-IgSF has an I-set fold topology that most closely resembles an I1-set domain (Fig 5C). However, β-IgSF lacks cysteines and disulphide bridges that are characteristic for I-set folds. Furthermore, the β-IgSF domain

possesses a truncated *C* strand that is directly followed by 1.5-turn α-helix (Fig 5C). These features were not observed in the structurally characterised IgI1 domains in the DALI or PDBeFOLD databases that β-IgSF most closely resembled, including macrophage colony stimulating factor 1 (MCS-F) and intracellular adhesion molecules 3 (ICAM-3) (Figs 5D and Fig EV3A and B). Therefore, the topology of β-IgSF domain represents a previously unrecognised IgI fold subtype, denoted here as I3-set Ig (IgI3) domain. Accordingly, the domain in β will now be referred to as β-IgI3. The unique features of this domain are (a) the absence of cysteine residues, (b) absence of *C'* and *C''* strands and (c) a truncated *C* strand that is directly followed by a 1.5-turn α-helix. Of note, the unique β-IgI3 region stretching from the *C* to the *D* strand possesses protruding hydrophobic residues, such as F42, located between the *C* strand and the α-helix, L46 located in the α-helix, and V53 located in the *CD* loop (Fig 5E).

## Identification of interacting residues in the (β-IgI3)-(CEACAM1-N) complex

CEACAMs form their homophilic and heterophilic interactions through the *A'GFCC'C''* face of the V-set domain (Figs 5A and 6A). Bacterial ligands also bind to the *A'GFCC'C''* face (Virji *et al*, 1999; Villullas *et al*, 2007; Conners *et al*, 2008; Korotkova *et al*, 2008; Bonsor *et al*, 2018; Moonens *et al*, 2018; Brewer *et al*, 2019). The *H. pylori* ligand HopQ uses a coupled folding and binding mechanism (Bonsor *et al*, 2018), and simulated docking indicated that the *E. coli* ligand AfaE binds to the CEACAM dimerisation interface through the *BE* strands and *DE* loop of an incomplete Ig fold (Anderson *et al*, 2004; Korotkova *et al*, 2008).

The structure determination by X-ray crystallography (Fig 6B; Appendix Table S4) of the complex of β-IgI3 and CEACAM1-N at 3.25 Å resolution showed that β-IgI3 also binds to the *A'GFCC'C''* face of CEACAM1 through residues located in the *C* to *D* strand region including the α-helix (Figs 6C and EV3C). To identify the residues forming contact sites between β-IgI3 and CEACAM1-N within the co-crystal, we used the NCONT sub-program of CCP4 (Appendix Table S5). Figure 6B shows the interacting residues on the surfaces of the two molecules. Specifically, L46 in the α-helix of β-IgI3 interacts by van der Waals forces with F29 and L95 (distances 3.34 Å and 3.59 Å) located in the *C* and *G* strands of CEACAM1, respectively (Fig EV3D; Appendix Table S5). In addition, the protruding β-IgI3 residue V53 is within close contact of I91 (distance 3.36 Å, *F* strand), Q44 (distance 2.50 Å, *C* to *C'* loop) and S32 (distance 3.41 Å, *C* strand) of CEACAM1, D55 is within close contact of Y34 (distance 3.50 Å, *C* strand) and G41 (distance

**Figure 5. The Ig-like domain of β protein represents a new IgI fold subtype.**

A  Dimer co-crystal structure of β-IgSF (blue) and CEACAM1-N (CC1-N; red).

B  Molecular structure of β-IgSF, in which each β-strand is coloured differently. The 1.5-turn α-helix shown in grey is located between the *C* strand and the *D* strand.

C  Illustration of the secondary structure organisation of the V-set, C1-set, C2-set, I1-set, I2-set and I3-set domains. Each secondary structure is composed of varying combinations of the β-strands *A, A', B, C, C', C'', D, E, F, G*. Disulphide bridges between the *B* strand and *F* strand are represented as a black dashed lines. The C-set domains lack the *A'* and *C''* strands and are subtyped into C1-set and C2-set based on presence and absence of the *D* strand, respectively. The I-set domains lack *C''* strand and are subtyped into C1-set and C2-set based on presence and absence of the *D* strand, respectively. The IgSF fold from β protein has the I3-set domain, characterised by absence of disulphide bridge, presence of the *D* strand, and presence of a truncated *C* strand followed by a 1.5-turn α-helix.

D  Comparison of β-IgI3 (blue) with the most similar folds; the IgI1 domain of MCS-F (turquoise) and the IgI1 domain of ICAM-3 (magenta).

E  The molecular surface mesh of β-IgI3 residues showing protruding residues located between the *C* and *D* strands in blue.

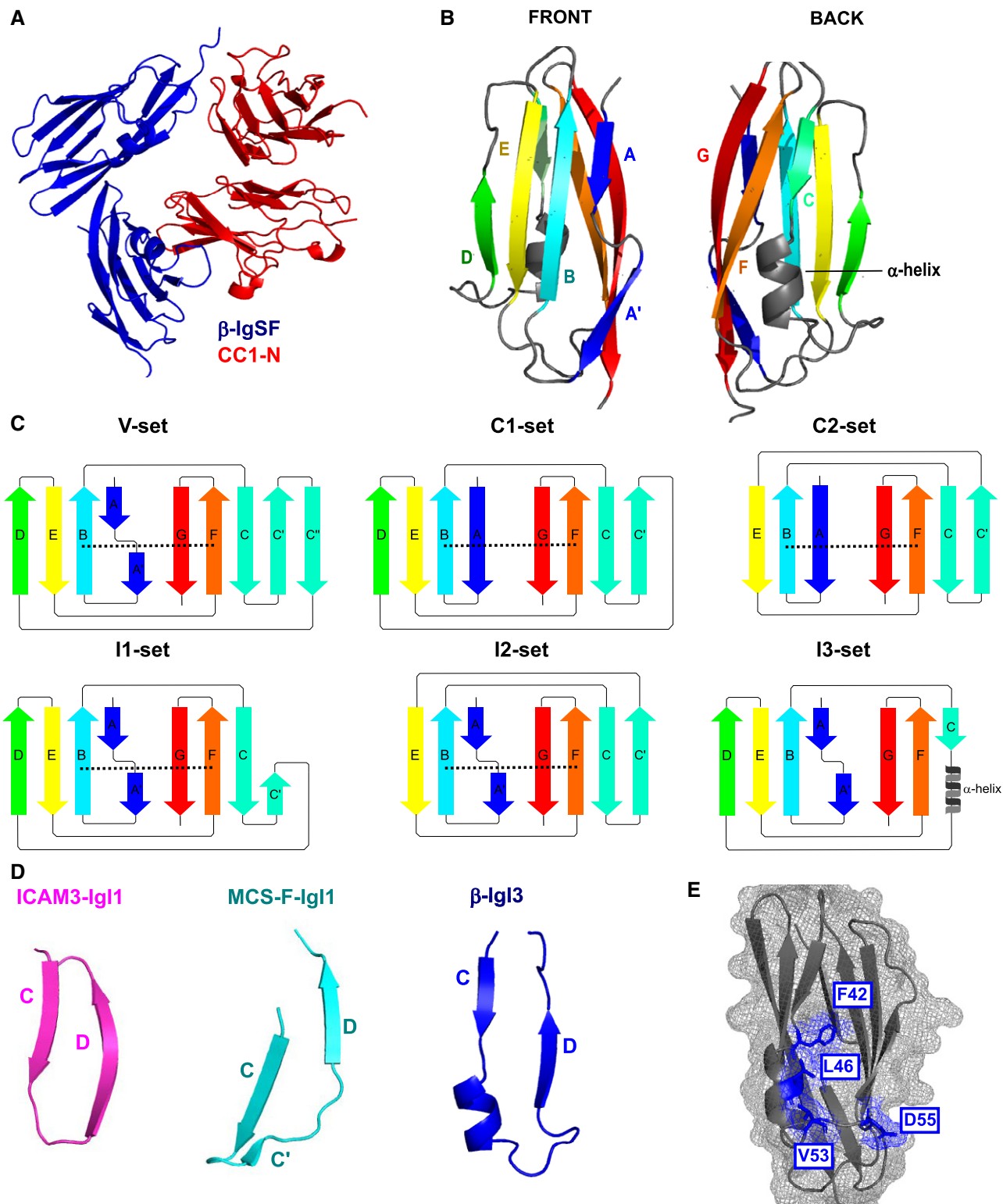

**Figure 5.**

2.90 Å, *C* to *C'* loop) of CEACAM1, and F42 is in close contact of L95 (distance 2.87 Å, *G* strand) of CEACAM1 (Appendix Table S5). Further contacts are made by β-IgI3 through residues L40, D41, S43, T47, N50, P51, S52, S54, I57, T59 and Y61 and CEACAM1 residues G30, Y31, Y34, D40, A49, T56, I91, S93, D94 and V96 of

CEACAM1, respectively (Appendix Table S5). No conformational changes were observed in CEACAM1 upon binding of β-IgI3 (Fig EV3E).

Based on the co-crystal data, we hypothesised that β-IgI3 residue L46 was critical for contacting CEACAM1-N via F29 and L95

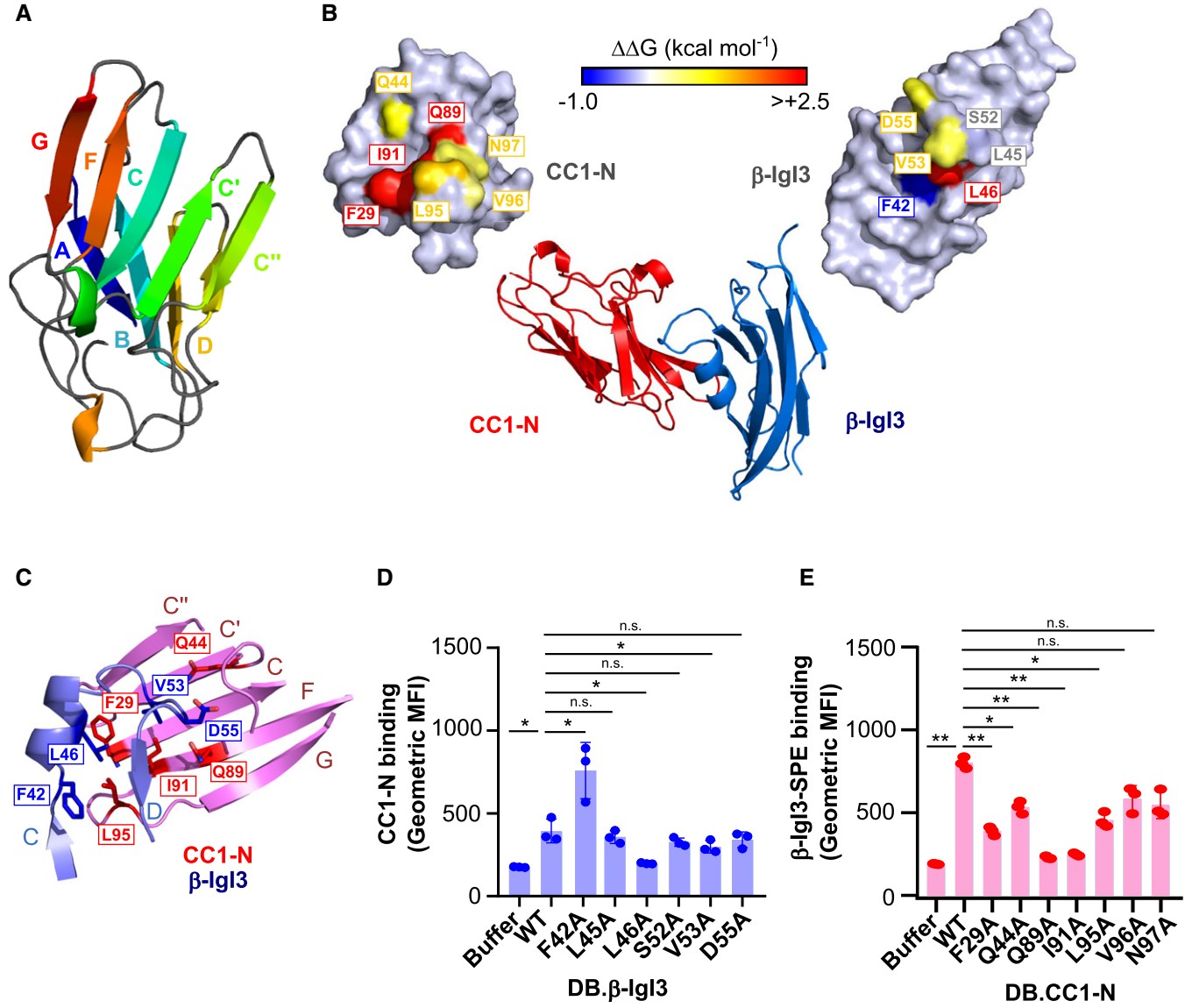

**Figure 6. The IgI3 domain of β protein binds CEACAM1 through the unique α-helix and *CD* loop.**

A  Molecular structure of CEACAM1 (CC1)-N, in which each strand of the IgV structure is coloured differently.

B  Molecular structure of the β-IgI3 and CC1 N-terminal domain co-complex. Residues that form the β-IgI3-CC1 interface are highlighted.

C  Close-up of the β-IgI3 (light blue) interface with CC1-N (pink), including highlighting of residues important for forming β-IgI3 (blue) and CC1-N (red) interactions. The *GFCC′C″* strands of CC1-N are shown and labelled, whilst the *C* to *D* strands of β-IgI3 are shown and labelled.

D  Binding of rCC1-N to dynabeads (DB) coated with β-IgI3 wild type (WT) or mutants. Mutation of β-IgI3 residues 42, 46 and 53 lead to significant changes in CC1-N binding capacity.

E  Binding of β-IgI3 coupled to Streptavidin-PE to DB coated with rCC1-N wild type or variants. Mutation of CC1-N residues 29, 44, 89, 91 and 95 lead to significant changes in β-IgI3 binding capacity.

Data information: Fluorescence of DB in (D and E) was measured by flow cytometry. Mean and SD values in (D and E) are displayed for $n = 3$ independent replicates. Statistical significance was calculated using $t$ test for (D and E), where $*P < 0.05$; $**P < 0.01$.

(Figs 6C and EV4B). Additionally, β-IgI3 residue V53 appeared critical for contacting CEACAM1-N residue I91, and β-IgI3 residue F42 was critical for contacting CEACAM1-N residue L95. We generated alanine mutations in β-IgI3 at these positions, and others, based on their contact with CEACAM1. ITC binding studies of these β-IgI3 mutants to "wild-type" unglycosylated CEACAM1-N revealed that the mutant β-IgI3^L46A failed to bind to rCEACAM1-N. Two additional

mutants, β-IgI3^V53A and β-IgI3^D55A, had reduced affinity to bind rCEACAM1-N ($K_D = 562 \pm 44$, $690 \pm 52$ nM, respectively; Fig EV4A; Appendix Table S6). In contrast, β-IgI3^F42A bound with higher affinity ($K_D = 16 \pm 15$ nM). We also tested the ability of unglycosylated rCEACAM1-N to bind DB coated with the β-IgI3 variants (Fig 6D). In this analysis, rCEACAM1-N displayed significantly reduced binding to DB coated with β-IgI3^L46A and β-IgI3^V53A and

significantly enhanced binding to DB coated with β-IgI3$^{F42A}$. Together, these data indicate that the L46, V53 and D55 residues of β-IgI3 are critical for the binding to CEACAM1. These residues were observed to be conserved in 57 β protein sequences (Appendix Fig S4).

For CEACAM1-N, the crystal structure of the complex indicated that β-IgI3 interacts with the dimer interface, contacting several residues which are important for CEACAM1 homodimerisation and for binding to other bacterial adhesins. We hypothesised that residues F29, I91 and L95 of CEACAM1 are critical for contacting β-IgI3 (Figs 6C and EV3C and D) and generated alanine mutants at these and several other positions in the homodimerisation interface. In ITC analysis with unglycosylated CEACAM1-N proteins, both CEACAM1-N$^{F29A}$ and CEACAM1-N$^{I91A}$ failed to bind β-IgI3 (Fig EV4B; Appendix Table S6). Additionally, CEACAM1-N$^{Q89A}$ also lacked ability to bind β-IgI3. Two mutants, CEACAM1-N$^{Q44A}$ and CEACAM1-N$^{L95A}$, resulted in a 10-fold decrease in binding affinities ($K_D$ = 996 ± 116 and 1350 ± 460 nM, respectively) whilst two further mutants, CEACAM1-N$^{V96A}$ and CEACAM1-N$^{N97A}$, displayed only a modest decrease (4-fold) in binding affinities ($K_D$ = 370 ± 4 and 490 ± 120 nM, respectively). In addition, we tested the ability of β-IgI3 coupled to streptavidin to interact with DB coated with the unglycosylated wild-type or mutant rCEACAM1-N (Fig 6E). Binding was significantly reduced for CEACAM1-N$^{F29A}$, CEACAM1-N$^{Q89A}$ and CEACAM1-N$^{I91A}$, as well as for CEACAM1-N$^{Q44A}$ and CEACAM1-N$^{L95A}$, confirming that residues F29, Q89 and I91 are major targets of β-IgI3 binding. I91 has also been identified as a critical CEACAM1 residue for interaction with *M. catarrhalis* (Conners *et al*, 2008), *Neisseria* spp. (Virji *et al*, 1999; Villullas *et al*, 2007), *H. influenzae* (Hill *et al*, 2001) and *Fusobacterium* spp (Wang, 2013). Though Q89 in CEACAM1 is critical for interaction with β-IgI3 and other bacterial ligands, Q89 was not in close contact with any β-IgI3 residues. It is possible that mutation of the CEACAM1 residue Q89 to an alanine forms a cavity on the surface that I91 and surrounding residues of CEACAM1 attempt to fill though alternative rotamer conformations that subsequently prevents interaction with β-IgI3. In summary, contact of residue L46 in β-IgI3 with CEACAM1 residue F29, and residue V53 in β-IgI3 with CEACAM1 residue I91, provide the critical interactions. Additional stability is gained through β-IgI3 residues F42 and D55.

### Comparison of β-IgI3- and HopQ-bound CEACAM1 structures

Comparison of the (β-IgI3)-(CEACAM1-N) complex and the (HopQ)-(CEACAM1-N) complex, revealed that the same set of CEACAM1 residues are contacted by both β-IgI3 and HopQ (Fig EV3F). However, HopQ is structurally completely unrelated to β-IgI3 (Bonsor *et al*, 2018). HopQ uses an intrinsically disordered loop, that folds into a β hairpin and a small helix upon binding to CEACAM1. The β hairpin extends the CEACAM1 *A′GFCC′C″* face whilst the small helix straddles the face (Fig EV3F). In contrast, β-IgI3 interacts through the small α helix and loop, with a single residue, L46, critical for β-IgI3 to fit into a pocket on CEACAM1-N formed by F29 and I91 (Fig 6C). Mutation of any of these residues in β-IgI3 or CEACAM1-N causes a hole in the protein–protein interface or collapse of the pocket, respectively, abolishing binding as observed with our ITC experiments.

### β-IgI3 homologs are broadly distributed in human Gram-positive bacteria

To gain a broader perspective on the possible role of the unique β-IgI3 structure in bacteria–host interactions, we investigated the distribution of this domain in the bacterial kingdom through BLAST analysis. A sequence similar or related to β-IgI3 was identified in 296 proteins predicted to be expressed by human Gram-positive bacteria, including several human pathogens, such as *S. pyogenes*, *S. dysgalactiae* and *S. pneumoniae*. These sequences formed several different clades upon maximum likelihood analysis (Fig 7A). The β-IgI3 domain was present in a large clade, clade I, which also included sequences from *S. oralis*. A second major clade, clade II, contained sequences from GBS and from other human streptococcal pathogens including *S. pyogenes* (also known as group A *Streptococcus*), *S. dysgalactiae* and *S. mitis*. Additionally, sequences closely related to clade II were detected in *S. mitis*, *S. milleri* and *S. intermedius*, forming clade III. Finally, many homologs of β-IgI3 were present in proteins from a diverse panel of Gram-positive human pathogens including *S. pneumoniae*, *S. pseudopneumoniae* and *Gemella haemolysans*. An additional homolog was identified in *Gardnerella vaginalis,* a Gram-variable bacterium. The sequences of β-IgI3 homologs were all located in proteins predicted to be surface-localised and anchored to the bacterial cell wall.

We predicted the tertiary structure of 11 representative homolog sequences based on the β-IgI3 crystal structure. These structures maintained the overall IgI3 fold, including an α-helix and loop located between the truncated *C* strand and the *D* strand (Appendix Fig S5), except clade XV sequence from *G. vaginalis* that lacked the *C* strand. These data suggest that the IgI3 structure is broadly distributed in bacterial cell wall-anchored adhesins. We focused further analysis on clade II that maintains key IgI3 structures whilst sharing 40% amino acid identity with β-IgI3 (Fig 7B; Appendix Fig S6A–C). This clade includes the R28 protein, which is found in *S. pyogenes* and in GBS (where the same protein is also called Alp3). R28 is a cell wall-anchored adhesin that is unrelated to β protein except for the IgI3 domain (Stålhammar-Carlemalm *et al*, 1999; Lindahl *et al*, 2005). To understand the CEACAM1 binding capacity of the putative IgI3 domain in R28, we inspected whether the critical protruding residues located between *C* and *D* strands in β-IgI3 (Fig 7C) were conserved in the IgI3 domain of R28. Though the CEACAM1-binding residues of β-IgI3 (F42, L46, V53 and D55) are not conserved in R28-IgI3 (Fig 7C), two of the R28-IgI3 residues found at corresponding positions were of hydrophobic nature as in β-IgI3. Interestingly, additional residue variation was evident amongst other IgI3 domains (Fig EV5). Since this result suggested that IgI3 homologs are likely to differ in their properties, it became critical to analyse whether the R28-IgI3 homolog binds CEACAM1.

### Human CEACAMs bind Gram-positive bacteria expressing β-IgI3 homologs

To determine whether the IgI3 domain in R28 interacts with CEACAM1, we purified rR28-IgI3 protein domain, that was expressed in *E. coli,* and tested interaction with rCEACAM1. DB coated with biotinylated R28-IgI3 and β-IgI3, but not HSA, interacted with rCEACAM1 (Fig 7D). In the reverse assay, DB coated with

rCEACAM1 interacted with R28-IgI3 and β-IgI3, but not HSA, coupled to streptavidin (Fig 7E). We also confirmed that rCEACAM1 bound to an R28-expressing *S. pyogenes* strain (Fig 7F). In addition, rCEACAM1 bound to an Alp3-expressing strain of GBS, which was expected given that R28 and Alp3 proteins are identical in sequence. To ascertain whether R28-IgI3 and CEACAM1 interactions were

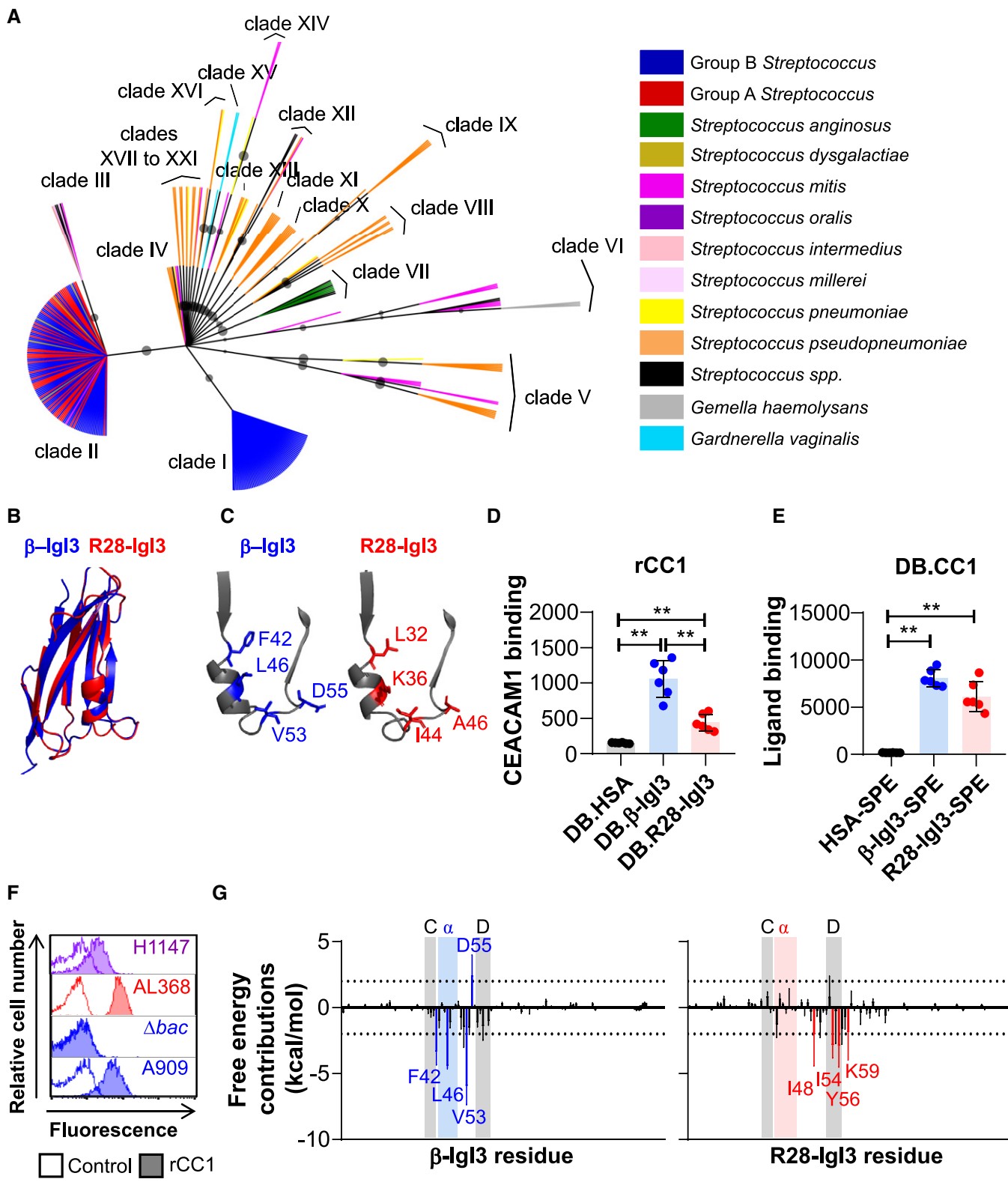

Figure 7.

◀

**Figure 7. The unique IgI3 structure is present in Gram-positive bacterial species and evidence for binding CEACAM1.**

A   Amino acid based maximum likelihood tree of β-IgI3 homologs. Clades of related IgI3 domains are shown, with clade I representative of β-IgI3 domain sequences. Branches are coloured by bacterial species from which the sequence originates, except those with no species designated that are shown in black. Maximum likelihood bootstrap values > 60 are shown as a circle, where larger circles indicate a higher bootstrap support value.

B   Superposition of the IgI3 domain from β protein (blue) onto predicted IgI3 domain structure of clade II sequence of *S. pyogenes* R28 protein (red).

C   Close-up of the truncated *C* strand, α-helix and *D* strand of β-IgI3 (blue) and R28-IgI3 (red), indicating the corresponding R28-IgI3 residues present at sites in β-IgI3 that bind CEACAM1 (CC1).

D   Binding of recombinant (r)CC1-HIS to dynabeads (DB) coated with β-IgI3, R28-IgI3 or control. Mean and SD is shown for *n* = 6 independent replicates.

E   Binding of β-IgI3, R28-IgI3 or human serum albumin (HSA) coupled to PE-conjugated streptavidin (SPE), to DB coated with CC1 (DB.CC1). Mean and SD is shown for *n* = 6 independent replicates.

F   Binding of recombinant rCC1-HIS (10 μg/ml) to R28-expressing GBS strain H1147 and R28-expressing *S. pyogenes* strain AL368. Data representative of *n* = 3 replicates.

G   The per residue free energy contribution spectrums of β-IgI3/CC1-N interface and R28-IgI3/CC1-N interface. The residues with binding free energy contributions lower than −2.0 kcal/mol are identified as key (favourable) residues and coloured blue (for β-IgI3) or red (for R28-IgI3). Free energy contributions at −2.0 kcal/mol and 2.0 kcal/mol are indicated by dashed lines. The position of the truncated *C* strand, α-helix and *D* strand is shaded for each protein. The mean and standard deviation is representative of estimates from 50 docked models.

Data information: Statistical significance was calculated using a one-way ANOVA with Sidak's multiple comparisons for (D and E), where **, *P* < 0.01. Fluorescence of DB (D and E) and bacteria (F) was measured by flow cytometry.

mediated through the N-terminal domain of CEACAM1, we used ITC to measure the affinity of R28-IgI3 for unglycosylated N-terminal domains of CEACAM1 (Appendix Fig S7). In this analysis, R28-IgI3 bound with intermediate affinity ($K_D$ = 1050 ± 18 nM, $\Delta H$ = −4.8 ± 0.3 kcal/mol) to CEACAM1-N. Taken together with the structural predictions described above, these data indicate that IgI3 folds present in adhesins from a wider range of Gram-positive bacterial pathogens can interact with the N-terminal domain of human CEACAM1, but that individual IgI3 domains may interact with human CEACAM1 differently.

To dissect whether β-IgI3 and R28-IgI3 interact differently with CEACAM1, we simulated the docking of IgI3 domains onto human CEACAM1 and examined the free energy calculations to ascertain the key IgI3 residues in the complex formation (Appendix Fig S6C). Residues with binding free energy contributions lower than −2.0 kcal/mol and greater than 2 kcal/mol are identified as key residues and unfavourable residues, respectively. Simulation of β-IgI3 and CEACAM1 docking correctly predicted that residues F42, L46 and V53 were key determinants of CEACAM1 binding, whilst D55 was an unfavourable determinant (Fig 7G; Appendix Fig S6D). The binding free energy of the (R28-IgI3)-(CEACAM1-N) complexes was strong (−31.02 ± 6.70 kcal/mol, n = 50 simulations). In contrast to β-IgI3, residues located in the α-helix of R28-IgI3 were not identified as key determinants of CEACAM1 binding (Fig 7G; Appendix Fig S6E). Instead, critical R28-IgI3 residues (K48, I54, I56 and K59) were located in the *CD* loop and the *D* strand only. Mapping of key residues on the surface structure therefore suggested that β-IgI3 and R28-IgI3 target CEACAM1 through different faces on the IgI3 fold (Appendix Fig S6D and E). Thus, IgI3 domains in different bacterial proteins probably interact with human CEACAM1 through mechanisms that are not identical but use the same structural fold.

## Discussion

We show here that a novel Ig-like fold in different streptococcal adhesins promotes binding to human CEACAM receptors. Molecular analysis of this domain in the β protein from GBS demonstrated that the Ig-like domain binds to the N-terminal domain of human CEACAM1 and CEACAM5. We solved the crystal structure of the Ig-like domain in β and revealed that it represents a novel Ig-fold structure related to the I-set that is characterised by an absence of cysteine residues and the presence of a truncated *C* strand that is directly followed by a unique 1.5-turn α-helix. As this Ig-like fold of the I-set has unique features, it was designated IgI3. Whilst cysteine residues and disulphide bridges are found in Ig folds of prokaryotic and eukaryotic proteins (Halaby & Mornon, 1998; Bodelón *et al*, 2013), the absence of cysteine residues has not been documented in IgI folds to date. In addition, the partial replacement of the *C* strand with a 1.5-turn α-helix was not identified in any other Ig fold in a search of PDB using HHpred (Zimmermann *et al*, 2018). I-set folds are rarely found within bacterial proteins as observed through a search of the SMART database (Letunic & Bork, 2018). This work therefore presents the first representative structure of a novel Ig-fold subtype, IgI3, and highlights that unique subtypes of Ig folds remain to be discovered in biology.

The co-crystal structure of the complex demonstrated that β-IgI3 targets the dimerisation interface of CEACAM1 through a unique mechanism involving residues in the *C* strand to *D* strand loop of β-IgI3 including the α-helix. In contrast, Opa proteins of *Neisseria* spp. are membrane proteins that bind to the CEACAM dimerisation interface through two hypervariable loops (Billker *et al*, 2000), UspA1 of *M. catarrhalis* forms a trimeric coiled-coil that binds to CEACAM dimerisation interface through a small bend (Conners *et al*, 2008), HopQ of *H. pylori* uses a disordered loop to bind the CEACAM dimerisation interface (Bonsor *et al*, 2018), and AfaE of *E. coli* uses an incomplete Ig-like domain to target the CEACAM dimerisation interface (Anderson *et al*, 2004; Korotkova *et al*, 2008).

The finding that β-IgI3 homologs are broadly distributed in Gram-positive bacteria raises many fascinating questions about the evolutionary history of IgI3-folds and their role in bacterial pathogenesis. What is the origin of the IgI3-fold? How conserved is the IgI3-fold structure? How did the IgI3-fold become so widely distributed? Do all bacterial adhesins containing IgI3 homologs bind CEACAMs and how is binding conferred? Whilst IgI3 homologs were found in putative adhesins from several Gram-positive bacteria, there was low conservation of the critical β-IgI3 residues in the predicted structures of homologs, suggesting that each sequence may possess different properties. Our study demonstrated that R28-IgI3 and β-IgI3 both bind the N-terminal domain of CEACAM1, though stimulated docking suggests that these IgI3 domains bind through alternative mechanisms. As R28-IgI3-like sequences were found in proteins from multiple human pathogens, CEACAM-

binding may represent an unrecognised adhesion strategy for a range of Gram-positive bacteria. Of note, the gene encoding β is encoded on mobile genetic elements that can move between bacteria by horizontal gene transfer (Tettelin *et al*, 2005). This could provide opportunities for rapid acquisition of CEACAM binding in other bacterial species leading to the emergence of strains with enhanced colonisation properties or virulence. Thus, the exploration of IgI3-fold and CEACAM receptor interactions is further warranted.

Adhesion to epithelial cells at mucosal surfaces is a prerequisite for streptococcal colonisation and disease development. For GBS bacteria, the data presented here show that the IgI3 domain in the β protein promotes binding to CEACAM1 and CEACAM5 present on epithelial cell surfaces. This finding is consistent with reports of CEACAM-interacting Gram-negative bacteria (Virji *et al*, 1996; Bos *et al*, 1998; Tchoupa *et al*, 2014, 2015; Gutbier *et al*, 2015; Königer *et al*, 2016; Javaheri *et al*, 2016). This is of general importance as it expands the spectrum of cellular adhesion mechanisms that may be utilised by GBS. Whilst the currently characterised interactions include the ability of the α adhesin to bind α1β1-integrin, BspC to bind vimentin and the binding of BspA to gp340 (Bolduc & Madoff, 2007; Rego *et al*, 2016; Deng *et al*, 2019), our demonstration that the β protein binds to CEACAM receptors identifies a novel type of interaction that potentially has important implications for pathogenesis. It is unclear whether the different GBS adhesins operate independently or in synergy with one another. Conceivably, they operate in synergy with adhesins that bind ECM components, such as Srr1, Srr2, FsbA, FsbB, FsbC, PilA or SfbA. As GBS can invade and survive within epithelial cells (Patras *et al*, 2015), we speculate that CEACAM engagement may also enhance virulence by promoting cellular invasion, as reported for CEACAM-binding proteins of Gram-negative pathogens (Billker *et al*, 2002; Korotkova *et al*, 2008; Tchoupa *et al*, 2015; Islam *et al*, 2018; Behrens *et al*, 2020). In addition, the possibility that binding to human CEACAMs mediated by β could modify epithelial cell responses and promote translocation of virulence factors, as described for *H. pylori* (Königer *et al*, 2016; Javaheri *et al*, 2016; Behrens *et al*, 2020), requires further investigation.

The capacity of individual pathogens to cause invasive systemic infections is dependent on their capacity to evade and/or subvert the host immune response. β protein interferes with antibody opsonisation by binding IgA and disrupting interaction with FcαR (Pleass *et al*, 2001) and interferes with complement opsonisation by binding factor H, which acts as a cofactor in the degradation of C3b (Areschoug *et al*, 2002b). Moreover, interactions with the inhibitory receptors Siglec-5 and Siglec-7, which are expressed on leucocytes, could enhance the immunosuppressive effects of β through induction of tyrosine phosphorylation (Carlin *et al*, 2009; Fong *et al*, 2018). As CEACAM1 is an inhibitory receptor, the binding of β might also induce tyrosine phosphorylation and CEACAM1-dependent immunosuppression, as been reported for Opa of *Neisseria* spp., UspA1 of *M. catarrhalis* and HopQ of *H. pylori* (Boulton & Gray-Owen, 2002; Slevogt *et al*, 2008; Javaheri *et al*, 2016; Gur *et al*, 2019). Thus, dual- or multi-engagement of β with different human ligands may have potent immunosuppressive effects. However, dissecting how β affects immune escape by interacting with different ligands will be challenging, particularly because the CEACAM and Siglec receptors have different cellular expression profiles. Moreover, the human specificity of the known ligand interactions presents further challenges for characterising the function of β

protein *in vivo*. Nonetheless, the data collectively suggest that β protein may act as a multitool protein that assists epithelial cell adhesion, cellular invasion and immunosuppression. Whilst not all strains of GBS express β, this protein is commonly expressed by strains of serotypes Ia, Ib, II and V (Lindahl *et al*, 2005) and it has been reported that expression of β at a high level is associated with increased virulence (Nagano *et al*, 2006). It therefore seems likely that β plays an important role in a large proportion of the serious infections caused by GBS, through its ability to bind CEACAMs and other human ligands.

In summary, our data demonstrate that an adhesin containing a domain with unique Ig-like fold, the IgI3 fold, promotes binding of GBS to human CEACAM1 and CEACAM5 receptors. Moreover, homologous domains were identified in a variety of predicted surface proteins from Gram-positive bacterial pathogens, suggesting that many Gram-positive bacterial pathogens employ IgI3 domains to engage human CEACAMs. Characterisation of these domains may lead to a better understanding of pathogenicity mechanisms and could pave the way for the development of novel therapeutic approaches to prevent infections.

# Materials and Methods

### Bacterial growth conditions

Organisms used in this study are listed in Appendix Table S1. Bacteria were cultured in Todd-Hewitt Broth (TH; *S. agalactiae*), TH Broth + 1% yeast (*S. pyogenes*), Tryptic Soy Broth (TSB; *S. aureus*) or Lysogeny Broth (LB; *E. coli*) and supplemented with antibiotics as listed in Appendix Table S1.

### Expression and purification of glycosylated and unglycosylated rCEACAMs

Glycosylated CEACAMs were expressed and purified from Expi293F cells (Life Technologies). In brief, gBlocks containing open reading frames (ORFs) encoding the extracellular domain of CEACAMs, and a C-terminal LPETGGS-6xHis tag, were cloned into the pcDNA3.4 expression vector (Invitrogen) (Appendix Table S2). Recombinant CEACAM-His proteins were expressed in Expi293F cells (Life Technologies) cultured in Expi293 Expression Medium (Life Technologies) and purified by affinity chromatography (ÄKTA Pure, GE Healthcare Life Sciences) using a Nickel column (GE Healthcare Life Sciences) as previously described (Zhao *et al*, 2020). Eluate was dialysed against 300 mM NaCl 50 mM Tris pH 7.8 at 4°C. Unglycosylated CEACAMs were expressed and purified from *E. coli* cultures. In brief, gBlocks containing ORFs encoding the N-terminal or A1B1A2 domains of CEACAM1, and a C-terminal LPETGGS-6xHis tag, were cloned into the pRSET-C vector (Appendix Table S2). Vectors were transformed into the Rosetta Gami (RG) *E. coli* strain. RG strains were cultured in LB supplemented with 100 µg/ml ampicillin and 1 mM D-glucose at 37°C overnight and subcultured 1:20 into LB supplemented with 1 mM D-glucose at 37°C until $OD_{600} = 0.4$. Protein expression was induced by culturing for 4 h at 37°C following the addition of 1 mM IPTG. After lysis of bacteria, proteins were purified using a Nickel column (GE Healthcare Life Sciences) and affinity chromatography (ÄKTA Pure, GE Healthcare

Life Sciences). Eluates were dialysed against 300 mM NaCl 50 mM Tris pH 7.8 at 4°C. Large-scale expression of tag-less CEACAM1-N domain and point mutants for use in isothermal titration calorimetry (ITC) and crystallisation were expressed using pET21d vectors in *E. coli* and purified as previously described (Bonsor *et al*, 2015a).

**Expression and purification of bacterial proteins**

β, α and Rib proteins were purified from GBS cultures as previously described (Lindahl *et al*, 1990; Stålhammar-Carlemalm *et al*, 1993). Domains of the β proteins (B6N, IgABR, B6C, IgSF and β75KN) and R28 (R28-IgI3) were cloned and expressed in *E. coli* (Appendix Table S2). The IgSF domain structure was found to be in a region overlapping the previously published B6C (amino acids 226–434) and β75KN domains (amino acids 434–788). (Areschoug *et al*, 2002b; Lindahl *et al*, 2005) Therefore, we have updated domain designations as follows: B6C represents amino acids 226–390, IgSF represents amino acids 391–503 and β75KN represents amino acids 504–788. Briefly, gBlocks containing ORFs and a C-terminal LPETGGS-6xHis tag were cloned into the pRSET-C vector using BamHI and NdeI. Vectors were transformed into the RG strain. RG strains were cultured, induced, harvested and lysed as described above for unglycosylated CEACAMs. Bacterial supernatants were filtered, supplemented with 10 mM imidazole and passed over a Nickel column (GE Healthcare Life Sciences). Proteins were purified by affinity chromatography (ÄKTA Pure, GE Healthcare Life Sciences), eluted with a gradient of imidazole (10–500 mM) and were dialysed against 300 mM NaCl 50 mM Tris pH 7.8 at 4°C using 6–8 kDa dialysis tubing.

**Binding of glycosylated and non-glycosylated recombinant CEACAM to bacteria**

$6 \times 10^6$ of mid-logarithmic phase bacteria were incubated with rCEACAM (1, 3, 5, 6 or 8), rCEACAM1-N or rCEACAM1-A1B1A2 at 4°C for 1 h with shaking. After washing in PBS + 0.1% bovine serum albumin (BSA), rCEACAM on the bacterial surface was detected by incubation with FITC-conjugated anti-HIS monoclonal antibody (mAb) at 4°C for 1 h with shaking. Bacterial fluorescence was measured by flow cytometric analysis following washing and fixation in 1% paraformaldehyde (PFA). To detect rCEACAM binding to GBS using pull-down Western blot analysis, bacterial pellets were resuspended in 1× sample buffer and heated to 95°C for 10 min. Lysates were separated by SDS-PAGE in a 12.5% polyacrylamide gel at 270V, blotted onto nitrocellulose membranes, blocked with 5% milk solution and probed with anti-CEACAM1 (clone C51X/8) mAb (BB Singer, Essen, Germany). Membranes were probed using rabbit anti-mouse-IgG-HRP and developed using ECL substrate.

**Measurement of β protein expression by GBS strains**

Rabbit antiserum was raised against β protein as previously described (Lindahl *et al*, 1990). $6 \times 10^6$ of mid-logarithmic phase bacteria were incubated with heat-inactivated 0.1% rabbit anti-β serum or normal rabbit serum at 4°C for 1 h with shaking, washed and incubated in the presence of PE-conjugated goat anti-rabbit-IgG at 4°C for 1 h with shaking. Fluorescence of bacteria was measured by flow cytometric analysis after washing and fixation in 1% PFA.

**Detection of CEACAM1 and β protein interaction**

***Western blot analysis***
Purified GBS proteins, or β protein domains (B6N, IgABR, B6C, IgSF or β75KN), were separated by non-reducing SDS-PAGE, transferred to nitrocellulose membranes, blocked with 5% milk solution overnight at 4°C, probed with 10 µg/ml rCEACAM1-His (4°C) and horse radish peroxidase (HRP)-conjugated mouse anti-His-IgG and developed using Pierce ECL Western Blot Substrate Reagent (Thermo Fisher Scientific).

***Binding of bacterial proteins to CEACAM1-coated dynabeads***
rCEACAM-His was attached to nickel dynabeads (DB; Dynabead His-tag pull-down and isolation, Invitrogen) following standard protocols. Presence of CEACAM1 was confirmed by incubating DB in the presence of 5 µg/ml anti-CEACAM or isotype control mAb, and detection with PE-conjugated goat anti-mouse-IgG mAb. To detect binding of purified GBS proteins, 40 µg of DB was incubated in the presence of 10 µg/ml of β protein and probed with 0.1% mouse antiserum or normal mouse serum, and secondary goat anti-mouse-IgG-PE mAb. Biotinylated proteins (including β-IgSF-biotin or HSA-biotin amongst others) were incubated with PE-conjugated Streptavidin on ice for 10 min. This generates biotin–streptavidin complexes on the remaining 3 binding pockets of streptavidin. To detect binding of proteins coupled with streptavidin (β-IgSF-biotin or HSA-biotin formed with Streptavidin-PE), 20 µg of DB was incubated with varying concentrations of protein–streptavidin. All dilutions and DB washes employed PBS + 0.1% BSA, all incubations were performed for 1 h at 4°C, and all fixations used 1% PFA. Fluorescence of DB was measured by flow cytometry. For inhibition experiments, CEACAM-coated DB were pre-incubated with 5 µg/ml anti-CEACAM1-N (clone CC1/3/5-Sab, LeukoCom, Essen, Germany), anti-CEACAM1-A1B1 mAb (clone B3-17, BB Singer, Essen, Germany) or 10 µg/mL rHopQ for 1 h at 4°C.

***Binding of CEACAM to bacterial protein coated dynabeads***
Proteins with C-terminal biotin tags were attached to streptavidin-coated dynabeads (Dynabead M-280 Streptavidin, Invitrogen) following standard procedures. To detect binding of rCEACAM-His or rCEACAM-N-His, $3 \times 10^5$ DB was incubated in the presence of 9 µl of proteins (concentrations range 0.1–30 µg/ml) and probed with FITC-conjugated mouse anti-His-IgG. In the case of rSiglec-5-Fc, binding was detected using anti-IgG-AF488. All dilutions and DB washes employed PBS + 0.1% BSA, all incubations were performed for 1 h at 4°C, and all fixations used 1% PFA. Fluorescence of DB was measured by flow cytometry.

**Isothermal titration calorimetry**

Duplicate ITC measurements were performed on an iTC200 instrument (GE Healthcare). Typically, 500 µM of unglycosylated CEACAM-N domains were loaded into the syringe of the calorimeter and 50 µM β-IgC$_2$ protein was loaded into the syringe. All measurements were performed at 25°C, with a stirring speed of 750 rpm, in 30 mM Tris-HCl, 150 mM NaCl, pH 7.5. Data were analysed using the Origin 7.0 software.

### Adhesion of GBS to epithelial cells

HeLa and Chinese Hamster Ovary (CHO) cell lines expressing CEACAMs were cultured as previously described (Daniel *et al*, 1993; Bos *et al*, 1998; Hollandsworth *et al*, 2020). Cells were seeded into 24-well tissue culture plates. Tightly confluent monolayers were infected with mid-logarithmic phase (absorbance at $OD_{600} = 0.4$) GBS strains at a multiplicity of infection (MOI) of 10. Assays were commenced by centrifugation for 2 min at 700 rpm and incubated at 37°C with 5% $CO_2$ for 30 min. Cells were washed five times with DMEM, detached with 0.25% trypsin and lysed (PBS + 0.025% Triton-X). Adherent bacteria were enumerated using serial dilution plating and growth on Todd-Hewitt agar plates. In specific assays, cells were pre-incubated with 5 µg/ml mAb (anti-CEACAM1-N or anti-CEACAM1-A1B1A2 mAb) for 60 min, or bacteria were pre-incubated with rCEACAM1-N for 60 min before co-culture. For confocal microscopy, experiments were performed with FITC-labelled GBS at a MOI of 10 for 30 min. After washing five times with PBS, monolayers were fixed with 4% PFA overnight prior to staining of nuclei with DaPI (Sigma Aldrich) and cell membranes with AF647-conjugated wheat germ agglutinin (Thermo Fisher Scientific). To perform assays with detached CHO cell lines, adherent monolayers were detached by incubation at 37°C for 5 min with Accutase Cell Detachment Solution (BioLegend). After washing, cells were resuspended to $3 \times 10^6$ cells/ml in DMEM containing 10% FCS. To measure CEACAM expression, cells were incubated with mouse mAb clone CC1/3/5-Sab (detects N-terminal domain of human CC1, CC3 and CC5) for 30 min, washed and incubated with PE-conjugated rabbit anti-mouse-IgG. To measure GBS binding, cells were incubated with FITC-labelled GBS at a MOI of 10 for 30 min at 4°C. After washing with PBS, cells were fixed in 4% PFA and fluorescence was measured by flow cytometry.

An overview of all cell lines used in this study is provided in Appendix Table S7. An overview of all antibodies used in this study is provided in Appendix Table S8.

### Crystallography of unglycosylated CEACAM1-N and β-IgSF

Initially, the β-IgSF and CEACAM1-N complex was prepared by mixing the proteins with a slight excess of CEACAM1-N (1:1.1 ratio). Protein was concentrated using an amicon Ultra-4 10,000 NMWL before purification by size exclusion chromatography using a Superdex 200 column (GE Healthcare) equilibrated with 20 mM Tris, 150 mM NaCl, pH 7.5. The complex was concentrated to 9 mg/ml and screened against the commercial Wizards I/II/III/IV (Rigaku) and JCSG$^+$ (QIAgen) screens using a Crystal Gryphon Protein Crystallography System (Art Robbins Instruments). Crystals were found in several conditions but diffraction was never found beyond 8.0 Å in the best condition of 1.8 M Ammonium Sulphate and 0.1 M Sodium Citrate pH 5.5. β-IgSF contains both a C-terminal Sortase and 6xHis tag. This was removed by first incubating β-IgSF by itself with Carboxypeptidase A (Sigma). 2 ml of 6 mg/ml of β-IgSF was incubated with 1:100 w/w of Carboxypeptidase A for 24 h at 4°C before the addition of another 1:100 w/w of Carboxypeptidase A and incubation for 24 h at room temperature. The reaction was terminated by the addition of EDTA to a final concentration of 5 mM. The protein was concentrated using an Amicon Ultra-4

10,000 NMWL before purification by size exclusion chromatography using a Superdex 200 column (GE Healthcare) equilibrated with 20 mM Tris, 150 mM NaCl, pH 7.5. CEACAM1-N was added in a slight excess (1:1.05 ratio) and purified by size exclusion as described above. The complex was concentrated and screened again at a concentration of 7.5 mg/ml. Crystals again formed in the condition of 1.8 M Ammonium Sulphate and 0.1 M Sodium Citrate pH 5.5; however, their morphology was different. Crystals were cyroprotected in 30 % v/v glycerol. A dataset was collected at beamline 12-2 at the Stanford Synchrotron Radiation Lightsource. Data were processed, merged, scaled and converted to structure factors using Aimless. Matthew's coefficient suggests five complexes in the asymmetric unit. Molecular replacement using MOLREP and CEACAM1-N (PDB entry 2gk2) as a search model in MOLREP only found two CEACAM1 molecules. Two copies of the complex would suggest a high solvent content (78 %). Refinement using REFMAC5 and visualisation in Coot showed that CEACAM1-N was well-placed and residual electron density was present for the β-IgI3 protein. Density modification was performed using Parrot (solvent flattening, histogram matching and NCS averaging) to improve the maps. The β-IgSF backbone was built manually. A PDBeFold search of the backbone showed that it was an IgI-like fold, with the closest fold by R.M.S.D being the IgC2 domain of Perlecan (PDB entry 1gl4). The Ig fold of β-IgSF possesses a previous unrecognised modification (Fig 5B and C). We denote this as the I3-set domain and the domain within β protein as β-IgI3. This was used to successfully position sidechains, lock the registry and provide extra restraints during refinement in REFMAC5 with ProSMART. The (β-IgI3)-(CEACAM1-N) complex has been deposited to the PDB with the entry code, 6V3P. Atom contacts in the (β-IgI3)-(CEACAM1-N) complex interface were identified using NCONT (CCP4) with a cut-off of 4.0 Å (Winn *et al*, 2011).

### Bioinformatics analysis

BLAST analysis of the β-IgI3 amino acid sequence was performed to identify homologs in bacterial proteins and subsequently aligned using ClustalW. Phylogenetic analysis was performed using maximum likelihood approach and 1,000 bootstrap replications in MEGA (Kumar *et al*, 2016), and the resulting tree displayed with interactive tree of life version 4 (Letunic & Bork, 2019). The structure of IgI3 homologs was predicted using the β-IgI3 structure as input in SWISS-MODEL (Webb & Sali, 2016). Only models passing a QMEAN score of $< -4.00$ were further analysed. Docking of β-IgI3 structure or the R28-IgI3 predicted structure to CEACAM1-N was simulated 50 times using ZDOCK server (Pierce *et al*, 2014), in which CEACAM1-N residues F29, Q44, Q89, I91 and N95 were included as contact sites. The free energy contribution of each simulation was interpedently calculated using MM/GBSA analysis on the HAWKDOCK server (Weng *et al*, 2019).

## Data availability

The datasets produced in this study are available in the following databases:

- Protein interaction (β-IgI3)-(CEACAM1-N) data: PDB 6V3P (https://www.rcsb.org/structure/unreleased/6V3P).

• The raw diffraction data have been deposited to the Integrated Resource for Reproducibility in Macromolecular Crystallography (https://proteindiffraction.org/).

All other data supporting the findings of this study are available within the paper and its supplementary information files and are available from the corresponding author on reasonable request.

**Expanded View** for this article is available online.

## Acknowledgments

The authors thank Carla J.C. de Haas, Piet Aerts, Kok P.M. van Kessel (UMC Utrecht, Utrecht, The Netherlands), Birgit Maranca-Hüwel, Bärbel Gobs-Hevelke (University of Duisburg-Essen, Germany) and Margaretha Stålhammar-Carlemalm (Lund University, Sweden) for excellent technical support and invaluable help. We thank Paul Sullam (University of California, USA) for providing *S. sanguinis* strains. We also wish to thank the support staff of Beamlines of 12-2 and ID23-D at the Stanford Synchrotron Radiation Light-source and the Advanced Photon Source for their aid in data collection, respectively. We thank Irene M. Mavridis (Institute of Child Health, Greece) and José R. Penadés (Imperial College London, UK) for critical reading of the manuscript. This work was supported by the European Union's Horizon 2020 Research and Innovation Programme under Grant Agreement 700862 (to A.J.M, J.A.G), Deutsche Forschungsgemeinschaft DFG grant SI-1558/3-1 (to B.B.S.), The Swedish Research Council grant K2011-56X-09490-21-6(to G.L), The Foundation Olle Engkvist Byggmästare (to G.L.) and the National Institutes of Health R01 NS116716 (to K.S.D.)

## Author contributions

Conception and design of the work (NMS, LD, DAB, EJS, JAGS, KSD, BBS, GL, AJM). Acquisition, analysis or interpretation of data (NMS, LD, DAB, JB, VS, ML, AS, ORN, EB, EL, JAGS, KSD, BBS, GL, AJM). Drafting of manuscript (NMS, LD, DAB, JAGS, KSD, BBS, GL, AJM). All authors read and commented on the final version of the manuscript.

## Conflict of interest

The authors declare that they have no conflict of interest.

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
