## [Review Process File · The EMBO Journal]

Bacterial protein domains with a novel Ig-like fold bind human CEACAM receptors

Nina van Sorge, Daniel Bonsor, Liwen Deng, Erik Lindahl, Verena Schmitt, Mykola Lyndin, Alexej Schmidt, Olof Nilsson, Jaime Brizuela, Elena Boero, Eric Sundberg, Jos van Strijp, Kelly Doran, Bernhard Singer, Gunnar Lindahl, and Alex McCarthy

DOI: [10.15252/embj.2020106103](https://doi.org/10.15252/embj.2020106103)

Corresponding authors: Alex McCarthy (a.mccarthy@imperial.ac.uk) , Gunnar Lindahl (gunnar.lindahl@med.lu.se)

Review Timeline:

Submission Date:	30th Jun 20
Editorial Decision:	26th Aug 20
Revision Received:	22nd Oct 20
Editorial Decision:	4th Dec 20
Revision Received:	8th Dec 20
Accepted:	10th Dec 20

Editor: Ieva Gailite

Transaction Report:

Thank you for submitting your manuscript for consideration by The EMBO Journal. I sincerely apologise for the protracted review process of your manuscript caused by delays in review submission. We have now received three referee reports on your manuscript, which are included below for your information.

As you will see from the comments, the reviewers appreciate the study, but they also indicate a number of issues that would have to be addressed and clarified before they can support publication of the manuscript. In particular, reviewer #1 with expertise in structural biology raises substantive concerns regarding the resolution and refinement of the structure, addressing of which will be essential. Furthermore, both reviewer #1 and reviewer #2 with expertise in pneumococcus infection mechanisms find that further insights into the functional relevance of the β protein/CEACAM interaction would have to be provided and indicate that the references to the existing literature have to be substantially improved. Based on the general interest expressed by the reviewers, I would invite you to address the concerns raised by the reviewers in a revised manuscript, especially focusing on the points outlined above.

I should add that it is The EMBO Journal policy to allow only a single major round of revision and that it is therefore important to resolve the main concerns at this stage. We are aware that many laboratories cannot function at full efficiency during the current COVID-19/SARS-CoV-2 pandemic, and I would be happy to discuss the revision in more detail via email or phone/videoconferencing.

We have extended our 'scooping protection policy' beyond the usual 3 month revision timeline to cover the period required for a full revision to address the essential experimental issues. This means that competing manuscripts published during revision period will not negatively impact on our assessment of the conceptual advance presented by your study. Please contact me if you see a paper with related content published elsewhere to discuss the appropriate course of action.

Please feel free to contact me if you have any further questions regarding the revision. Thank you for the opportunity to consider your work for publication. I look forward to receiving the revised manuscript.

Referee #1:

The manuscript "Bacterial protein domains with a novel Ig-like fold target human CEACAM receptors" by Sorge et al identifies an interaction between the beta protein of group B streptococci and CEACAM receptors, show this as a high affinity interaction, determine the co-crystal structure (no easy feat!), and show that this is important for binding to epithelial cells. This is a novel finding and augments the known interactions between GBS and potential human host receptors. The data are compelling that this is a strong interaction.

Major concerns

The rationale for this study is framed in terms of relevance to GBS associated infections, but the data do not demonstrate that this particular interaction is relevant to any of the GSB-related infections stated in the introduction. They writing over-implies that this interaction contributes to

infection without evidence that this is so. In fact, other GBS adhesive proteins are shown to be critical for virulence in the infections listed in the introduction, and the references and discussion of these is modest at best or absent (for example, a notable absence in the references is a reference to the meningitis-associated interaction between GBS Srr1 and Srr2 and fibrinogen from Seo et al (2012) PLoS Pathog. 8, e1002947 and follow up work, which clearly shows that this is related to virulence). It is unclear whether this adhesin works in synergy with others... or whether this is a virulence factor at all, versus an interaction important for commensalism... which would make more sense given the binding to epithelial cells. The manuscript would benefit from a more thorough analysis that included appropriate consideration of the literature. As a note, the majority of the current discussion section repeats the findings of the results and has limited value. The discussion could instead be used to place the findings within the context of the field. Finally, there are many places where the wording is imprecise to the point of being factually inaccurate (or difficult to understand). Some of these are listed below, but I am sure that I have missed some, and it will be important for the senior authors to read this carefully for accuracy.

It is not clear how the computational structure shown in a main text figure relates to the experimental structure. Or why this computational structure (which is incorrect) is shown given the presence of an experimental structure.

Crystal structure

I posit that the majority of crystallographers would say that the resolution is not what the authors state; I certainly disagree with the cutoff. An R_{sym} of >9 (I didn't realize that the program could output a number that high) and R_{sym} of >3 with I/σ of <1 seems very poor. Even if one only uses $CC1/2$ for the cutoff and doesn't take a holistic view of what all of the statistics are telling you about the data quality, this is on the edge. Sometimes in situation, authors may report in the table legend the resolution where I/σ is >2 of R_{sym}/R_{pim} is < 0.5 , which are more traditional cutoffs. Of concern, there are no electron density figures that allow a reader to evaluate whether the quality of the maps is consistent with ~ 3 Å resolution. The PDB validation report shows refinement statistics that are reasonable but not outstanding for this resolution, which is unusual because these are compared to all structures in the PDB, including those determined decades ago, and modern refinement programs can usually get stats in the top 20-30% without trouble (the PDB validation report often does not correctly output RSRZ stats, so that is fine and not considered as a part of this evaluation).

Introduction.

Aspects of the introduction are not sufficient for a broad audience. As noted above, there is an unacceptable omission of what is known in the field for adhesive interactions to cells by GBS and which of these have been shown to be key for virulence. This follows with CECAMS, for example, in lines 93-97, a little more explanation detailing some of the findings in the citations listed would be nice. To further prepare this manuscript for a broader audience, a more general statement of how serotypes can affect adhesion and bacterial pathogenesis could go a long way. It would also be worth noting why it was expected that additional adhesive interactions might exist given what is already known.

Minor

1. Line 144, why specifically use CC1 to screen for interactions?
2. FigS1a, the positions of the arrows hide parts of the gel, specifically CC3 but perhaps all lanes. Can the bands be highlighted another way?
3. Lines 119-120: "Notably, rCEACAM1 only bound to GBS strains included in our screen (Fig. 1A).

Please reword. It sounds like you are saying that CC1 doesn't bind to strains of GBS that you didn't test.

4. Line 123, reads like you studied the paper instead of referencing the study.

5. Fig 1E legend, line 1032, please comment that this is validation of the antibody and not an experiment.

6. Line 153-154, reword. It sounds like all of the domains have an Ig fold.

7. Fig 2F, why do you think the sample with B-IgSF and anti-CC1-A1B1 signal is greater than for B-IgSF alone?

8. Line 182, please define IgC2 in the main text.

9. Fig 4C, could variation be due to something in the media, intercellular interactions or cell density? You stated earlier that certain domains in the CEACAMs facilitate hetero and homophilic interactions. Also, for supplementary figure 5A-B, why is the geometric mean for the CC5 cells similar to control when using anti-CC1 & CC5?

10. Fig 5G It is very difficult to see the residues.

11. Supplementary Fig 6B. Please report an RMSD for the superposition.

12. Supplementary table 5, Please add a citation for ccp4.

13. Supplementary Table 6, please report what the +/- values are.

14. Line 304, You state the mutant b-IgI3 L52A has a Kd of 234 nM. I could not find this data in supplementary table 6. Instead there is a b-IgI3 S52A mutant with a Kd of 85nM. Supplementary Fig 7A also shows an S52A mutant.

15. Line 335, Please elaborate on what you mean by "forms a gap".

16. Supplementary Fig 6C would be a little more clear if the HopQ gray was a little darker.

17. Line 374, please remove this if it is a question.

18. Line 375, "despite having only..." Most proteins with this level of identity will have the same fold, so this is not unexpected.

19. When discussing the presence of this Ig fold variant, could you comment on predictions of the conserved disulfide bond found in other Ig folds. To the best of my knowledge, these are only found in mammalian Ig folds and not in any bacterial proteins with Ig folds, so the absence of a disulfide is not surprising. Much is made of the slight variation of the I-set Ig fold, but slight variations of Ig folds are relatively common. This discussion could be shortened, and Fig. 5 moved to the supplement.

20. I would like to see a Kd from ITC experiments for the R28-IgI3 protein to CC1 to verify that it has a physiologically relevant affinity since the binding relative to B-IgI3 seems to vary between Fig7D and E.

21. It would be interesting to see a binding experiment between R28-IgI3 and the N-domain of CC1 to compare it to the full length CC1.

22. While you used ITC analysis of alanine mutations to confirm the residues that interact between B-IgI3 and CC1-N, I would like some analysis of the crystal contacts to additionally support this.

23. Line 499, bacteria not bacterial

24. Line 543, state the filter size.

25. Line 546, please state the imidazole concentration that the protein eluted at.

26. For western blot experimental procedures, please detail how the blots were blocked. Also, please add the company that the antibodies are from (line 558).

27. I would encourage the authors to deposit the raw crystallography data to an appropriate repository.

28. Fig 3 - it is not clear from the writing or the presentation that the ITC analyses were performed with replicate measurements (there are no error bars on the graphs).

29. Fig. S3 - I could not see error bars on the controls. If these are smaller than the symbol, perhaps this could be noted in the legend (or better yet, could each replicate measurement be shown as in Fig 6 and others?)

Referee #2:

This manuscript is an excellent structural biology analysis of B protein of Gp B streptococcus that indicates a domain with a previously unrecognized Ig fold promotes binding to host cell CEACAMs 1 and 5. The structure of the protein is described in detail with very strong data and clear illustrations. The role in adherence is documented by standard procedures with appropriate controls and mutant analysis. The extension of the structural analysis to other species with potential homologs is interesting.

While the structural biology is excellent, the biological insights or potential for function in pathogenesis are not discussed. While it could be argued that this is an area for a separate manuscript, it appears that there is a lot known about the function of this protein in other bacteria and this could then be readily tested to determine if the domain identified is of biological significance. This is particularly important since the role of many of the other domains in virulence is known. At a minimum, a more detailed description of the biology proposed for this protein would improve the understanding by the reader of what significance the interaction described might be.

Referee #3:

Nina M. van Sorge et al. report a novel broadly applied mechanism for bacteria to target human carcinoembryonic antigen-related cell adhesion molecules (CEACAM). The CEACAM recognition is based on the bacterial protein beta adhesin. The authors showed that protein beta binds to CEACAM1 and CEACAM5 receptors using its IgSF domain of a novel Ig-fold subtype. The authors revealed the molecular basis of interactions between this domain and CEACAMs, and demonstrated that the unique features of the novel sub-fold are involved in the binding. In my opinion, this paper is an important contribution to our understanding of host-bacterial pathogen interactions. It is well written and well illustrated. I have two suggestions regarding the structural part:

1. The asymmetric unit contains two complexes. Could the authors provide a superposition of the complexes? Are the interactions between molecules essentially the same in the complexes?
2. Superpositions in main figures 5 and 7 (as well as those in supplementary) should be shown as stereo views to be seen clearly.

Referee #1:

The manuscript "Bacterial protein domains with a novel Ig-like fold target human CEACAM receptors" by Sorge et al identifies an interaction between the beta protein of group B streptococci and CEACAM receptors, show this as a high affinity interaction, determine the co-crystal structure (no easy feat!), and show that this is important for binding to epithelial cells. This is a novel finding and augments the known interactions between GBS and potential human host receptors. The data are compelling that this is a strong interaction.

Major concerns

The rationale for this study is framed in terms of relevance to GBS associated infections, but the data do not demonstrate that this particular interaction is relevant to any of the GSB-related infections stated in the introduction. They writing over-implies that this interaction contributes to infection without evidence that this is so. In fact, other GBS adhesive proteins are shown to be critical for virulence in the infections listed in the introduction, and the references and discussion of these is modest at best or absent (for example, a notable absence in the references is a reference to the meningitis-associated interaction between GBS Srr1 and Srr2 and fibrinogen from Seo et al (2012) PLoS Pathog. 8, e1002947 and follow up work, which clearly shows that this is related to virulence). It is unclear whether this adhesin works in synergy with others... or whether this is a virulence factor at all, versus an interaction important for commensalism... which would make more sense given the binding to epithelial cells. The manuscript would benefit from a more thorough analysis that included appropriate consideration of the literature. As a note, the majority of the current discussion section repeats the findings of the results and has limited value. The discussion could instead be used to place the findings within the context of the field. Finally, there are many places where the wording is imprecise to the point of being factually inaccurate (or difficult to understand). Some of these are listed below, but I am sure that I have missed some, and it will be important for the senior authors to read this carefully for accuracy.

RESPONSE

We thank the reviewer for these comments and have modified the introduction and discussion to consider the biological consequences of these interactions for a broader audience. More specifically, we have now modified i) the introduction to clarify current knowledge about GBS adhesins, and ii) the discussion to provide a more thorough consideration of the potential impact of CEACAM-engagement on epithelial cells, immune cells and virulence. The changes we have made are detailed below.

Changes to Introduction:-

We have expanded the background to place the study into the broader context of the GBS field. We highlight the specific adhesins currently known to interact with extracellular matrix (ECM) components and cellular receptors.

Lines 78-98: "GBS interacts with several ECM constituents and the molecular mechanisms underpinning these interactions are well-defined. The adhesins currently known to interact with ECM components are the fibrinogen-binding adhesins Srr1, FsbA, FsbB and FsbC (Seo et al, 2012; Buscetta et al, 2014; Gutekunst et al, 2004; Schubert et al, 2002), keratin 4-binding adhesin Srr1 (Samen et al, 2007), fibronectin-binding adhesins BsaB and SfbA (Jiang & Wessels, 2014; Mu et al, 2014), and collagen-binding adhesin PilA (Banerjee et al, 2011). These interactions promote epithelial colonization (Sheen et al, 2011; Wang et al, 2014; Schubert et al, 2004; Samen et al, 2007)

as well as cellular invasion and/or invasive disease (Deng *et al*, 2019; Banerjee *et al*, 2011; Seo *et al*, 2012; Tenenbaum *et al*, 2005; Mu *et al*, 2014; van Sorge *et al*, 2009). Generally, our knowledge of the mechanisms that GBS utilizes to directly adhere to host cells is limited (Patras & Nizet, 2018; Bolduc & Madoff, 2007; Pietrocola *et al*, 2018); α adhesin binds $\alpha 1\beta 1$ -integrin to promote epithelial cell internalization (Bolduc & Madoff, 2007), vimentin-binding adhesin BspC (Deng *et al*, 2019), and BspA adhesin interacts with gp340, a mucin-like glycoprotein associated with the surface of mucosal tissues (Rego *et al*, 2016). However, the host receptor targets of several putative adhesins remain uncharacterized, including Rib, Sip, LrrG, HvgA and BibA proteins (Stålhammar-Carlemalm *et al*, 1993; Brodeur *et al*, 2000; Santi *et al*, 2007; Tazi *et al*, 2010). Consequently, it is expected that several GBS adhesin-host factor interactions remain uncharacterized. Their identification is important for development of a vaccine or anti-bacterial strategies that interfere with GBS mucosal colonization (Pietrocola *et al*, 2018; Larsson *et al*, 1996; Heath, 2016; Michel *et al*, 1992)."

Changes to Discussion:-

We have expanded the discussion of the biological consequences of CEACAM1 engagement to GBS cellular adhesion and colonisation.

Lines 494-514: "Adhesion to epithelial cells at mucosal surfaces is a prerequisite for streptococcal colonization and disease development. For GBS bacteria, the data presented here show that the IgI3 domain in the β protein promotes binding to CEACAM1 and CEACAM5 present on epithelial cell surfaces. This finding is consistent with reports of CEACAM-interacting Gram-negative bacteria (Bos *et al*, 1998; Gutbier *et al*, 2015; Tchoupa *et al*, 2014, 2015; Königer *et al*, 2016; Javaheri *et al*, 2016; Virji *et al*, 1996). This is of general importance as it expands the spectrum of cellular adhesion mechanisms that may be utilized by GBS. While the currently characterized interactions include the ability of the α adhesin to bind $\alpha 1\beta 1$ -integrin, BspC to bind vimentin and the binding of BspA to gp340 (Rego *et al*, 2016; Bolduc & Madoff, 2007; Deng *et al*, 2019), our demonstration that the β protein binds to CEACAM receptors identifies a novel type of interaction that potentially has important implications for pathogenesis. It is unclear whether the different GBS adhesins operate independently or in synergy with one another. Conceivably, they operate in synergy with adhesins that bind ECM components, such as Srr1, Srr2, FsbA, FsbB, FsbC, PilA or SfbA. As GBS can invade and survive within epithelial cells (Patras *et al*, 2015), we speculate that CEACAM engagement may also enhance virulence by promoting cellular invasion, as reported for CEACAM-binding proteins of Gram-negative pathogens (Islam *et al*, 2018; Billker *et al*, 2002; Korotkova *et al*, 2008; Behrens *et al*, 2020; Tchoupa *et al*, 2015). In addition, the possibility that binding to human CEACAMs mediated by β could modify epithelial cell responses and promote translocation of virulence factors, as described for *H. pylori* (Behrens *et al*, 2020; Königer *et al*, 2016; Javaheri *et al*, 2016), requires further investigation."

We have provided a thorough discussion of the role of β protein as a virulence factor and the impact of CEACAM1 engagement on immune responses.

Lines 512-534: "The capacity of individual pathogens to cause invasive systemic infections is dependent on their capacity to evade and/or subvert the host immune response. β protein interferes with antibody opsonization by binding IgA and disrupting interaction with Fc α R (Pleass *et al*, 2001) and interferes with complement opsonization by binding factor H, which acts as a cofactor in the degradation of surface bound C3b (Areschoug *et al*, 2002b). Moreover, interactions with the inhibitory receptors Siglec-5 and Siglec-7, which are expressed on leukocytes, could enhance the immunosuppressive effects of β through induction of tyrosine phosphorylation (Carlin *et al*, 2009; Fong *et al*, 2018). As CEACAM1 is an inhibitory receptor, the binding of β might also induce tyrosine phosphorylation and CEACAM1-dependent immunosuppression, as been reported for Opa of *Neisseria spp.*, UspA1 of *M. catarrhalis* and HopQ of *H. pylori* (Javaheri *et al*, 2016; Boulton & Gray-

Owen, 2002; Slevogt *et al*, 2008; Gur *et al*, 2019). Thus, dual- or multi-engagement of β with different human ligands may have potent immunosuppressive effects. However, dissecting how β affects immune escape by interacting with different ligands will be challenging, particularly because the CEACAM and Siglec receptors have different cellular expression profiles. Moreover, the human specificity of the known ligand interactions presents further challenges for characterizing the function of β protein *in vivo*. Nonetheless, the data collectively suggests that β protein may act as a multitool protein that assists epithelial cell adhesion, cellular invasion and immunosuppression. While not all strains of GBS express β , this protein is commonly expressed by strains of serotypes Ia, Ib, II and V (Lindahl *et al*, 2005) and it has been reported that expression of β at a high level is associated with increased virulence (Nagano *et al*, 2006). It therefore seems likely that β plays an important role in a large proportion of the serious infections caused by GBS, through its ability to bind CEACAMs and other human ligands.”

It is not clear how the computational structure shown in a main text figure relates to the experimental structure. Or why this computational structure (which is incorrect) is shown given the presence of an experimental structure.

RESPONSE

We have retained Fig 2E, as this was an important piece of preliminary analysis that led to the identification of the CEACAM1 binding site in β protein. We have modified text to

Lines 193-194: “We assessed whether the IgSF domain shared structural homology with other resolved IgSF structures.”

Crystal structure. I posit that the majority of crystallographers would say that the resolution is not what the authors state; I certainly disagree with the cutoff. An Rsym of >9 (I didn't realize that the program could output a number that high) and Rsym of >3 with I/sig of <1 seems very poor. Even if one only uses CC1/2 for the cutoff and doesn't take a holistic view of what all of the statistics are telling you about the data quality, this is on the edge. Sometimes in situation, authors may report in the table legend the resolution where I/sig is >2 or Rsym/Rpim is < 0.5, which are more traditional cutoffs. Of concern, there are no electron density figures that allow a reader to evaluate whether the quality of the maps is consistent with ~3 Å resolution. The PDB validation report shows refinement statistics that are reasonable but not outstanding for this resolution, which is unusual because these are compared to all structures in the PDB, including those determined decades ago, and modern refinement programs can usually get stats in the top 20-30% without trouble (the PDB validation report often does not correctly output RSRZ stats, so that is fine and not considered as a part of this evaluation).

RESPONSE

The data was truncated to 3.25Å, resulting in Rsym = 1.386 in the outer shell (Rpim = 0.635 accounting for redundancy/multiplicity and an I/sig of 2.2, and re-refined to an R/Rfree of 21.8/2.4 which is hopefully more acceptable. Material and Methods, Results, Table 1, Figures and Supplemental Data have been updated to reflect this. An electron density map of interface residues of β bound to CEACAM1 has also been added at Fig. EV3A.

Introduction.

Aspects of the introduction are not sufficient for a broad audience. As noted above, there is an unacceptable omission of what is known in the field for adhesive interactions to cells by GBS and

which of these have been shown to be key for virulence. This follows with CECAMS, for example, in lines 93-97, a little more explanation detailing some of the findings in the citations listed would be nice. To further prepare this manuscript for a broader audience, a more general statement of how serotypes can affect adhesion and bacterial pathogenesis could go a long way. It would also be worth noting why it was expected that additional adhesive interactions might exist given what is already known.

RESPONSE

We have provided a more detailed description of the known adhesins of GBS in the introduction as outlined above.

We have provide further detail on known bacterial ligands of human CEACAMs:-

Lines 114-121: “The characterized bacterial adhesins include Opa of *Neisseria spp.*, HopQ of *Helicobacter pylori*, UspA1 of *Moraxella catarrhalis*, P1 of *Haemophilus influenzae*, Dr adhesins of *Escherichia coli* and CbpF of *Fusobacterium spp.* Interaction of these adhesins with human CEACAMs promotes cellular adhesion, cellular invasion, translocation of virulence factors and tissue penetration (Islam *et al*, 2018; Königer *et al*, 2016; Tchoupa *et al*, 2015; Billker *et al*, 2002; Tchoupa *et al*, 2014; Korotkova *et al*, 2008; Javaheri *et al*, 2016). Of note, these CEACAM-binding bacterial adhesins are structurally distinct, implying that they arose through convergent evolution.”

We have provided a general statement on GBS serotypes and virulence:-

Lines 65-66: “From a total of 10 different serotypes, GBS belonging to serotypes Ia, Ib, II, III and V are most commonly associated with disease cases worldwide (Edmond *et al*, 2012).

As outlined above and as recently reviewed by Pietrocola *et al.* (Frontiers in Microbiology, 2018, 9; 602. DOI: 10.3389/fimmu.2018.00602), several adhesins reported to possess adhesive properties have no host factor binding partner characterized. We have clarified why previously unidentified host-pathogen interactions are likely to exist in the introduction as follows:-

Lines 94-98: “Consequently, it is expected that several GBS adhesin-host factor interactions remain uncharacterized. Their identification is important for development of a vaccine or anti-bacterial strategies that interfere with GBS mucosal colonization (Pietrocola *et al*, 2018; Larsson *et al*, 1996; Heath, 2016; Michel *et al*, 1992).”

Minor

1. Line 144, why specifically use CC1 to screen for interactions?

RESPONSE

We have modified to

Lines 144-145: “We used CEACAM1 in the screen as it binds all known bacterial ligands of CEACAMs”

2. FigS1a, the positions of the arrows hide parts of the gel, specifically CC3 but perhaps all lanes. Can the bands be highlighted another way?

RESPONSE

We have modified Fig. EV1A to highlight bands with dashed boxes rather than arrows.

3. Lines 119-120: "Notably, rCEACAM1 only bound to GBS strains included in our screen (Fig. 1A). Please reword. It sounds like you are saying that CC1 doesn't bind to strains of GBS that you didn't test.

RESPONSE

We have modified to

Lines 149-150: "Notably, rCEACAM1 only bound to the indicated GBS strains"

4. Line 123, reads like you studied the paper instead of referencing the study.

RESPONSE

We thank the reviewer for making this point. We have included more detail and modified to

Lines 152-158: "We utilized the GBS genome comparison dataset of Tettelin *et al.* (2005), which includes CEACAM1 binding (A909 and H36B) and non-binding (18RS21 and NCTC10/84) strains, to identify genes encoding cell-wall anchored proteins that were associated with rCEACAM1-binding phenotype. We identified that genomic island of diversity region 3.1 was present in rCEACAM1-binding strains (A909 and H36B) and absent in non-binding strains (515, COH1, NEM316 and NCTC10/84)."

5. Fig 1E legend, line 1032, please comment that this is validation of the antibody and not an experiment.

RESPONSE

This is a pull-down experiment that confirms binding of rCEACAM1 to the surface of GBS A909. We have clarified this in the legend "Pull-down of human rCC1-HIS (30 µg/mL) by GBS strain A909 strain analyzed by Western blot analysis."

6. Line 153-154, reword. It sounds like all of the domains have an Ig fold.

RESPONSE

Corrected.

7. Fig 2F, why do you think the sample with B-IgSF and anti-CC1-A1B1 signal is greater than for B-IgSF alone?

RESPONSE

It is possible that binding of mAb to the A1B1A2 domain reduces flexibility of CEACAM1 and thereby increases stability of the interaction between β-IgSF and the N-terminal domain of CEACAM1. As we currently have no firm evidence to support this hypothesis, we feel it is not appropriate to comment on this minor observation in this manuscript.

8. Line 182, please define IgC2 in the main text.

RESPONSE

Corrected to “IgSF”

9. Fig 4C, could variation be due to something in the media, intercellular interactions or cell density? You stated earlier that certain domains in the CEACAMs facilitate hetero and homophilic interactions. Also, for supplementary figure 5A-B, why is the geometric mean for the CC5 cells similar to control when using anti-CC1 & CC5?

RESPONSE

We have expanded on discussion of this point, with the following

Lines 253-256: “It is possible that the variation in GBS adhesion could be due to differences in CEACAM-expression density by CHO cells or due to differences in β expression by GBS strains in replicated experiments.”

We thank the reviewer for raising this issue. The flow cytometry data in the submission showed that CC5 is present on the surface of the CHO.CC5 cell line after treatment with Accutase, as mAb 5C8C4 (specific for CC5) can bind. However, we do not see binding of mAb CC1/3/5-Sab to the CHO.CC5 cell line. Currently, we cannot explain the reason, and therefore we have removed data for the CHO.CC5 cell line – specifically, data using CHO.CC5 has been removed from Fig 4E, 4F, EV2A and EV2B. Text has been modified accordingly.

10. Fig 5G It is very difficult to see the residues.

RESPONSE

We have made Fig 5G larger and modified the colours to improve visualisation of the residues.

11. Supplementary Fig 6B. Please report an RMSD for the superposition.

RESPONSE

The RMSD is now reported in the Figure legend.

12. Supplementary table 5, Please add a citation for ccp4.

RESPONSE

Added.

13. Supplementary Table 6, please report what the +/- values are.

RESPONSE

Values are averages of duplicate runs with their associated error. Text has been modified.

14. Line 304, You state the mutant b-IgI3 L52A has a Kd of 234 nM. I could not find this data in supplementary table 6. Instead there is a b-IgI3 S52A mutant with a Kd of 85nM. Supplementary Fig 7A also shows an S52A mutant.

RESPONSE

We agree this is a mistake in the text. Data in Appendix Table S6 is correct and shows S52A mutant has an affinity of 85nM.

Lines 339-341: "Two additional mutants, β -IgI3^{V53A} and β -IgI3^{D55A}, had reduced affinity to bind rCEACAM1-N ($K_D = 562 \pm 44, 690 \pm 52$ nM, respectively) (Fig. EV4A; Appendix Table S6)."

15. Line 335, Please elaborate on what you mean by "forms a gap".

RESPONSE

We have added additional clarity

Lines 368-371: "It is possible that mutation of the CEACAM1 residue Q89 to an alanine forms a cavity on the surface that I91 and surrounding residues of CEACAM1 attempts to fill though alternative rotomer conformations that subsequently prevents its interaction with in β -IgI3."

16. Supplementary Fig 6C would be a little more clear if the HopQ gray was a little darker.

RESPONSE

Modified as suggested.

17. Line 374, please remove this if it is a question.

RESPONSE

Corrected.

18. Line 375, "despite having only..." Most proteins with this level of identity will have the same fold, so this is not unexpected.

RESPONSE

We have modified to

Lines 406-408: "We focused further analysis on clade II that maintains key IgI3 structures whilst sharing 40% amino acid identity with β -IgI3 (Fig 7B; Appendix Fig. S5A, S5B & S5C)."

19. When discussing the presence of this Ig fold variant, could you comment on predictions of the conserved disulfide bond found in other Ig folds. To the best of my knowledge, these are only found in mammalian Ig folds and not in any bacterial proteins with Ig folds, so the absence of a disulfide is not surprising. Much is made of the slight variation of the I-set Ig fold, but slight variations of Ig folds are relatively common. This discussion could be shortened, and Fig. 5 moved to the supplement.

RESPONSE

Bacterial Ig folds can contain cysteine residues and disulfide bridges. We have included a sentence in the discussion to highlight this point

Lines 460-461: “While cysteine residues and disulphide bridges are found in Ig folds of prokaryotic and eukaryotic proteins (Halaby & Mornon, 1998), the absence of cysteine residues has not been documented in IgI folds to date.”

We believe that the absence of C' strand and presence of a truncated C strand are not slight variations of I-set Ig fold. These features mean the Ig fold structure is not represented by the IgI1 nor IgI2 folds, as displayed in Fig 5C. Therefore, the description of a new IgI fold structure is an important finding, and consequently we believe Fig. 5 is best placed as a main Figure and not as EV or Appendix Figures.

20. I would like to see a Kd from ITC experiments for the R28-IgI3 protein to CC1 to verify that it has a physiologically relevant affinity since the binding relative to B-IgI3 seems to vary between Fig7D and E.

RESPONSE

We have now performed ITC analysis to confirm and quantify interaction between R28-IgI3 and CC1, as shown below, which has a kD of 1050 ± 18 nM. This Figure is included as Appendix Fig. S6. We have modified the text in results and discussion as below:-

Lines 428-432: “To ascertain whether R28-IgI3 and CEACAM1 interactions were mediated through the N-terminal domain of CEACAM1, we used ITC to measure the affinity of β -IgSF for unglycosylated N-terminal domains of CEACAM1 (Appendix Fig. S6). In this analysis, R28-IgI3 bound with intermediate affinity ($K_D = 1050 \pm 18$ nM, $\Delta H = -4.8 \pm 0.3$ kcal mol^{-1}) to CEACAM1-N.”

Lines 484-486: “Our study demonstrated that R28-IgI3 and β -IgI3 both bind the N-terminal domain of CEACAM1, though stimulated docking suggests that these IgI3 domains bind through alternative mechanisms.”

21. It would be interesting to see a binding experiment between R28-IgI3 and the N-domain of CC1 to compare it to the full length CC1.

RESPONSE

The data in Fig. 7D, 7E and 7F indicated that R28-IgI3 interacts with full-length glycosylated CEACAM1. Our new data from ITC experiments in Appendix Fig. S6 demonstrate that R28-IgI3 binds to the N-terminal domain of CEACAM1. Collectively, this indicates that R28-IgI3 interacts with the N-

terminal domain of CEACAM1 through a protein-protein interaction. Work is now in progress to characterize the molecular details of this interaction and contribution of glycans.

22. While you used ITC analysis of alanine mutations to confirm the residues that interact between B-IgI3 and CC1-N, I would like some analysis of the crystal contacts to additionally support this.

RESPONSE

We previously analysed the crystal contacts through NCONT analysis provided in Appendix Table S5 and described in results. In order to clarify this, we have modified the results as follows:-

Lines 319-321: "To identify the residues forming contact sites between β -IgI3 and CEACAM1-N within the co-crystal, we used the NCONT sub-program of CCP4 (Appendix Table S5)."

23. Line 499, bacteria not bacterial

RESPONSE

Corrected

24. Line 543, state the filter size.

RESPONSE

Corrected.

25. Line 546, please state the imidazole concentration that the protein eluted at.

RESPONSE

Corrected.

26. For western blot experimental procedures, please detail how the blots were blocked. Also, please add the company that the antibodies are from (line 558).

RESPONSE

Corrected. Appendix Table S8 lists all antibodies used and manufactures.

27. I would encourage the authors to deposit the raw crystallography data to an appropriate repository.

RESPONSE

Raw data is now deposited as outlined in Data Availability section

Lines 724-726: "The X-ray diffraction dataset has been deposited to the Integrated Resource for Reproducibility in Macromolecular Crystallography (<https://proteindiffraction.org/>)."

28. Fig 3 - it is not clear from the writing or the presentation that the ITC analyses were performed with replicate measurements (there are no error bars on the graphs).

RESPONSE

Duplicate runs were performed. Each trace is a representative of one of the duplicate runs. Figure legends, Table legends and materials and methods text has been updated to state duplicate experiments were conducted.

29. Fig. S3 - I could not see error bars on the controls. If these are smaller than the symbol, perhaps this could be noted in the legend (or better yet, could each replicate measurement be shown as in Fig 6 and others?)

RESPONSE

We have added clarity into the legend "Note: error bars in controls are smaller than symbols."

Referee #2:

This manuscript is an excellent structural biology analysis of β protein of Gp B streptococcus that indicates a domain with a previously unrecognized Ig fold promotes binding to host cell CEACAMs 1 and 5. The structure of the protein is described in detail with very strong data and clear illustrations. The role in adherence is documented by standard procedures with appropriate controls and mutant analysis. The extension of the structural analysis to other species with potential homologs is interesting.

While the structural biology is excellent, the biological insights or potential for function in pathogenesis are not discussed. While it could be argued that this is an area for a separate manuscript, it appears that there is a lot known about the function of this protein in other bacteria and this could then be readily tested to determine if the domain identified is of biological significance. This is particularly important since the role of many of the other domains in virulence is known. At a minimum, a more detailed description of the biology proposed for this protein would improve the understanding by the reader of what significance the interaction described might be.

RESPONSE

We have provided a more detailed discussion of the role of β protein as a virulence factor and the impact of CEACAM1 engagement on immune responses.

Lines 515-537: “The capacity of individual pathogens to cause invasive systemic infections is dependent on their capacity to evade and/or subvert the host immune response. β protein interferes with antibody opsonization by binding IgA and disrupting interaction with Fc α R (Pleass *et al*, 2001) and interferes with complement opsonization by binding factor H, which acts as a cofactor in the degradation of surface bound C3b (Areschoug *et al*, 2002b). Moreover, interactions with the inhibitory receptors Siglec-5 and Siglec-7, which are expressed on leukocytes, could enhance the immunosuppressive effects of beta through induction of tyrosine phosphorylation (Carlin *et al*, 2009; Fong *et al*, 2018). As CEACAM1 is an inhibitory receptor, the binding of β might also induce tyrosine phosphorylation and CEACAM1-dependent immunosuppression, as been reported for Opa of *Neisseria spp.*, UspA1 of *M. catarrhalis* and HopQ of *H. pylori* (Javaheri *et al*, 2016; Boulton & Gray-Owen, 2002; Slevogt *et al*, 2008; Gur *et al*, 2019). Thus, dual- or multi-engagement of β with different human ligands may have potent immunosuppressive effects. However, dissecting how β affects immune escape by interacting with different ligands will be challenging, particularly because the CEACAM and Siglec receptors have different cellular expression profiles. Moreover, the human specificity of the known ligand interactions presents further challenges for characterizing the function of β protein *in vivo*. Nonetheless, the data collectively suggests that β protein may act as a multitool protein that assists epithelial cell adhesion, cellular invasion and immunosuppression. While not all strains of GBS express β , this protein is commonly expressed by strains of serotypes Ia, Ib, II and V (Lindahl *et al*, 2005) and it has been reported that expression of β at a high level is associated with increased virulence (Nagano *et al*, 2006). It therefore seems likely that β plays an important role in a large proportion of the serious infections caused by GBS, through its ability to bind CEACAMs and other human ligands.”

Referee #3:

Nina M. van Sorge et al. report a novel broadly applied mechanism for bacteria to target human carcinoembryonic antigen-related cell adhesion molecules (CEACAM). The CEACAM recognition is based on the bacterial protein beta adhesin. The authors showed that protein beta binds to CEACAM1 and CEACAM5 receptors using its IgSF domain of a novel Ig-fold subtype. The authors revealed the molecular basis of interactions between this domain and CEACAMs, and demonstrated that the unique features of the novel sub-fold are involved in the binding. In my opinion, this paper is an important contribution to our understanding of host-bacterial pathogen interactions. It is well written and well illustrated. I have two suggestions regarding the structural part:

1. The asymmetric unit contains two complexes. Could the authors provide a superposition of the complexes? Are the interactions between molecules essentially the same in the complexes?

RESPONSE

RMSD of the two complexes is low (0.20Å) and no real differences are observed. Analysis was performed on the complex comprised of chain B and chain C. Text has been modified to address this when introducing structural data.

Lines 279-283: "The asymmetric unit contains two molecules of the (β -IgSF)-(CEACAM1-N) complex, which are similar to each other when superimposed (r.m.s.d 0.20Å). All analysis of the (β -IgSF)-(CEACAM1-N) complex has been performed with the chains B and C of the co-crystal complex."

2. Superpositions in main figures 5 and 7 (as well as those in supplementary) should be shown as stereo views to be seen clearly.

RESPONSE

Superposition figures (Fig. 5D, 5E, EV3C) have been converted to stereo, and their legends updated. As addition of stereo views to Appendix Fig. 4 will make the figure extremely complex, we prefer to keep this figure unchanged.

Dear Alex,

Thank you for submitting a revised version of your manuscript. Your revised study has now been assessed by one of the original referees, who finds that most of their main concerns have been addressed, but also notes some remaining issues that still have to be clarified before they can recommend publication of the manuscript. Therefore, I would like to invite you to address the remaining referee comments and the following editorial issues:

1. Our publisher has done their pre-publication check on your manuscript. I have attached the file here. Please take a look at the word file and the comments regarding the figure legends and respond to the issues. Please also use this version when you resubmit the revised version.
2. Our editorial guidelines do not allow inclusion of "data not shown" references in the manuscript - currently such can be found in lines 237 and 352. Please include the missing data on beta-IgSF domain conservation in GBS proteins in the Appendix.
3. Please move "References" section above "Figure Legends".

Please let me know if you have any further questions regarding any of these points. You can use the link below to upload the revised files.

Referee #1:

The authors responses directly address the overwhelming majority of the concerns, and the manuscript is much improved, particularly in terms of accessibility to a broad readership and accuracy of statements that summarize literature in the field. There remain minor concerns surrounding the crystal structure and some wording anomalies.

The authors mention in the response to reviewers that they have now truncated the resolution to 3.25A resolution, and have updated the table. However, the PDB validation report still indicates 2.95A resolution. It seems as if the PDB deposition needs to be updated.

The electron density in Fig. EV3 is not convincing - I am hoping that this is just presentation. This just looks like a blob and does not appear to be consistent with the accuracy that one would anticipate at ~3-3.5 A resolution. At these resolutions, one would expect that objects that are ~3-3.5 A apart would appear to be distinct and that side chain density would be consistent with the sequence. Neither is apparent in this figure panel.

Line 468-469 on disulfides in bacterial Ig folds. The cited reference, which is a >20 year old review, does not indicate in its text that disulfides are found in bacterial Ig folds. With a quick search, I do find non-canonical locations of disulfides within bacterial Ig folds. It would be worth identifying and referencing at least one explicit example to ensure that this statement is accurate.

Fig. 7 (title) "The unique Ig13 structure is widely present..." having something be both 'unique' and 'widely present' seems contradictory. Please revise wording.

Line 706 "Data was processed..." should be "Data were processed..."

Line 739, "The X-ray diffraction dataset..." I believe that the authors mean "The raw diffraction data..."

Reviewer Comments:

1. The authors mention in the response to reviewers that they have now truncated the resolution to 3.25Å resolution, and have updated the table. However, the PDB validation report still indicates 2.95Å resolution. It seems as if the PDB deposition needs to be updated.

RESPONSE

It appears the new PDB validation report was not uploaded upon resubmission. We have uploaded the new PDB validation report.

2. The electron density in Fig. EV3 is not convincing - I am hoping that this is just presentation. This just looks like a blob and does not appear to be consistent with the accuracy that one would anticipate at ~3-3.5 Å resolution. At these resolutions, one would expect that objects that are ~3-3.5 Å apart would appear to be distinct and that side chain density would be consistent with the sequence. Neither is apparent in this figure panel.

RESPONSE

The electron density may not appear to be consistent with the side chains is most likely due to a couple of factors; (i) the solvent content of the crystal is ~80%. (ii) B factors are much higher than normal. (iii) the data is anisotropic along the I axis ($I/\sigma = 2$ at 4.2Å).

We are confident in the placement of the side-chains/registry due to our mutagenesis data of the CEACAM1 and β -Ig13 interface residues. Furthermore, the same electron density effects are observed in CEACAM1 of our structure. For instance, the electron density of the G strand of CEACAM1 in our structure looks like a “blob” [Fig A], even though several higher resolution structure of dimeric CEACAM1 exist in the PDB and display normal electron density. We have updated Fig. EV3C to more clearly show electron density of β -Ig13.

Fig A: Electron density of the G strand of CEACAM1.

3. Line 468-469 on disulfides in bacterial Ig folds. The cited reference, which is a >20 year old review, does not indicate in its text that disulfides are found in bacterial Ig folds. With a quick search, I do find non-canonical locations of disulfides within bacterial Ig folds. It would be worth identifying and referencing at least one explicit example to ensure that this statement is accurate.

RESPONSE

We have now included citation of Bodelon *et al.* 2013. Immunoglobulin domains in *Escherichia coli* and other enterobacteria: from pathogenesis to applications in antibody technologies. *FEMS Microbiol Rev* 37(2):204-50. The review of Bodelon *et al.* discusses structure and variation of Ig folds in *E. coli* and other bacterial proteins. Citation included at line 461.

4. Fig. 7 (title) "The unique IgI3 structure is widely present..." having something be both 'unique' and 'widely present' seems contradictory. Please revise wording.

RESPONSE

We agree. Modified to "The unique IgI3 structure is present in Gram-positive bacterial species and evidence for binding CEACAM1." at line 1200.

5. Line 706 "Data was processed..." should be "Data were processed..."

RESPONSE

Line 692 - Modified as suggested.

6. Line 739, "The X-ray diffraction dataset..." I believe that the authors mean "The raw diffraction data..."

RESPONSE

Line 725 - Modified as suggested.

Editor accepted the manuscript.

Corresponding Author Name: Alex J McCarthy

Journal Submitted to: The EMBO Journal

Manuscript Number: EMBOJ-2020-106103